



# Detection and climatology of Saharan dust frequency and mass at the Jungfraujoch (3580 m asl, Switzerland)

Martine Collaud Coen[1], Benjamin Tobias Brem[2], Martin Gysel-Beer[2], Robin Modini[2], Stephan Henne[3], Martin Steinbacher[3], Davide Putero[4], Maria I. Gini[5], and Kostantinos Eleftheriadis[5]

[1]Federal Office of Meteorology and Climatology, MeteoSwiss, CH-1530 Payerne, Switzerland
[2]PSI Center for Energy and Environmental Sciences, 5232 Villigen PSI, Switzerland
[3]Swiss Federal Laboratories for Materials Science and Technology, Empa, CH-8600 Dübendorf, Switzerland
[4]National Research Council of Italy – Institute of Atmospheric Sciences and Climate, CNR–ISAC, Turin, Italy
[5]Environmental Radioactivity & Aerosol Tech. for Atmospheric & Climate Impacts, INRaSTES, National Centre of Scientific Research "Demokritos", Athens, Greece

**Correspondence:** Martine Collaud Coen, martine.collaudcoen@meteoswiss.ch

**Abstract.** Saharan dust (SD) can be transported over long distances by large-scale atmospheric circulation. SD events (SDE) occur 30 to 150 times each year at the high-altitude station of the Jungfraujoch (JFJ) in the Swiss Alps. The SD detection method, applied since 2001, is based on the inversion of the single scattering albedo wavelength dependence caused by the higher coarse-mode fraction and the chemical composition of dust. Here, the reproducibility of the SD detection by different types of nephelometers and absorption photometers is first investigated and is then compared to detections based on the observed concentration of coarse-mode aerosol, source sensitivities simulated with FLEXPART as well as to the dust index provided by the Copernicus Atmospheric Monitoring Service. The difference in SD detection is stronger for various nephelometer types than for various absorption photometers. Each detection method has advantages and weakness and no one can be considered as reference. The climatology of the 23-year time series of dust hours and dust mass at the JFJ shows that the temporal influence of dust is strongest from February to June, and in October and November, whereas the dust mass is higher in spring than in fall. The SDEs detected by a high coarse-mode particle concentration have different sources and pathways to Europe than the ones detected by the optical method. The inhomogeneity in the SD time series and the high inter-annual variability restrain the evaluation of long-term trends.

## 1 Introduction

In arid regions soil dust is released into the atmosphere as a result of wind erosion and are beside sea salt the main sources of the aerosol load in the Earth's atmosphere. The Sahara and Sahel regions of northern Africa are the largest sources of mineral dust particles world-wide, with 790-840 million tons per year (Ginoux et al., 2012), corresponding to around half of global dust emissions. The soil dust released from the Sahara desert is transported in all directions including the Atlantic Ocean, South America, the Caribbean, Southern Africa, the Middle East, the Mediterranean basin, Europe, and even East Asia (Liu et al., 2022), resulting in high spatial and temporal variability of the dust concentration over several tens of thousands of kilometers away from its source. Emission and transport processes both largely controlled by meteorological factors lead to a temporal




heterogeneity at multiple time scales (Querol et al., 2019). The presence of mineral dust in the atmosphere has a broad impact on radiative forcing through solar light extinction and cloud formation, as well as on air quality and subsequent adverse health effects. After deposition, soil dust acts as ocean and vegetation nutrient, but also increases snow and ice melting as a result
of decreased surface albedo, and has several repercussions on our technosphere, e.g., on photovoltaic energy production and aircraft engine wear (Mona et al., 2023).

Saharan dust (SD) concentrations can be very large in the Mediterranean basin with high impacts on air quality, the western part of Europe being more frequently subject to Saharan dust events (SDEs) than the eastern part (Gkikas et al., 2021). Moreover, while the most frequent strong SD episodes (7.3 episodes/year defined as dust optical depth larger than the mean aerosol
optical depth (AOD) plus four standard deviations, corresponding to 0.5-1.3 over the Mediterranean basin) are observed in the western Mediterranean basin (especially in SE Spain and the Alboran basin), extreme SD episodes occur more frequently (2.6 episodes/year) in the central and eastern Mediterranean basin (Gkikas et al., 2009). The impact of SD is less pronounced in central Europe and decreases further in the northern countries with a strong decrease in the contribution of SD to AOD at 40-47 °N (Marinou et al., 2017). SDE can be detected by their physical, optical, and chemical characteristics. Mineral dust
transported across Europe has mode diameters between 11 and 50 $\mu$m with decreasing size in northern Europe (Rostási et al., 2022). However, particles larger than 50-500 $\mu$m can also be transported over thousands of kilometers (Adebiyi et al., 2023; Varga et al., 2021; van der Does et al., 2018; Ryder et al., 2018). Consequently, SD often constitutes the predominant fraction of the coarse-mode in the free troposphere and they have usually a non-spherical shape, so that detection methods based on the concentration of coarse-mode aerosol and on the linear depolarization ratio of the back-scattered signal of LiDAR are often
used by in situ and remote sensing observations. The concentration and type (mostly hematite or geothite) of iron-oxide in SD particles determine their impact on the absorbed solar radiation and hence on the absorption coefficient, cross section and AOD (Sokolik and Toon, 1999; Di Biagio et al., 2017). They are mainly present in diameters smaller than 2 $\mu$m as pure or externally mixed Fe oxide hydroxides and tend to be internally mixed in larger particles (Panta et al., 2023). Long-range transport of the small iron-rich particles is therefore favored. Finally, chemical analysis of collected dust allows a precise determination of
particle mineralogy and shape (Rostási et al., 2022; Jeong, 2024), giving further information on the sources, mixing state, and aging of mineral dust.

SDEs time series are assessed by ground- or space-based remote sensing instrumentation, but also by in situ measurements of dust in the atmosphere, deposited on the ground, or washed out by precipitation (Mona et al., 2023). As a general rule (Moteki, 2023), off-line laboratory investigations are very precise, e.g., to determine the chemical composition and heterogeneity of
mineral dust, but they have very low temporal resolution and concern only a few sampling sites. In situ observations usually have a high temporal and spatial resolution of optical, physical, and chemical dust properties, but the short-lived nature of dust largely restricts their spatial representation. Remote sensing observations characterize the total atmospheric column, profiling techniques also providing a vertical resolution of the dust properties. Physical and optical properties can be provided by retrieval algorithms such as GRASP (Dubovik et al., 2021), which may allow distinction between several aerosol types, though
this requires apriori assumptions. Finally, satellite measurements excel at monitoring the spatial and temporal development of dust incursions at the expense of the characterization of the dust and the sensitivity to the surface layer. All of these detection



methods also suffer from some potential artifacts, such as the restriction of remote sensing observations in the presence of clouds or precipitations, detection limits for weak columnar AOD, repeat cycle of satellites or sampling and instrumental artifacts for in situ measurements.

Several studies use the Ångström exponents of optical properties (scattering, absorption, and extinction coefficients, as well as the single scattering albedo (SSA)) to characterize the average composition and size of the aerosol (e.g. Cappa et al., 2016; Schmeisser et al., 2017; Romano et al., 2019; Costabile et al., 2013). Fialho et al. (2005) and Sandradewi et al. (2008) used the absorption Ångström exponent to estimate the relative contribution of biomass burning and dust or of biomass burning and fossil fuel, respectively. Valentini et al. (2020) and Kaskaoutis et al. (2021) extended the use of optical properties to the

absorption spectral curvature and to the single scattering co-albedo Ångström exponent to further characterize the optical properties of various aerosol compounds. Collaud Coen et al. (2004) published a mineral dust detection method based on the SSA Ångström exponent (SSAAE) from aerosols in situ measurements, which was validated by backward trajectories (71% of cases) and the color of the TSP filter (74%). This methodology, which will be called the optical method in this paper, was initially applied to in situ measurements at the Jungfraujoch (JFJ) high-altitude research station for a 2 year period. It has

been used not only at other high-elevation sites such as Chacaltaya (Bolivia) (Chauvigné et al., 2019) and Sonnblick (Austria) (Schauer et al., 2016), but also at lower altitude background stations such as in Cyprus (Drinovec et al., 2020), Spain (at Granada, Segura et al. (2014), Montsec, Monsegny and Barcelona, (Ealo et al., 2016; Yus-Díez et al., 2021)), as well as in cities in Spain (Yus-Díez et al., 2021) and Italy (Valentini et al., 2020). Similar algorithms were also used to detect desert dust from passive remote sensing instrumentation (e.g. Valentini et al., 2020; Russell et al., 2010; Rupakheti et al., 2019; Ge et al.,

2010; Kim et al., 2005; Valenzuela et al., 2015) .

The present study reassesses the optical method (Collaud Coen et al., 2004) for different pairs of nephelometers and filter-based absorption photometers at JFJ and Mount Helmos (HAC), Greece, and compares them with other detection methods based on the aerosol size distribution, the simulated dust concentration of the Copernicus Atmosphere Monitoring Service (CAMS), source sensitivities calculated with the FLEXible PARTicle dispersion model (FLEXPART) and back-trajectories

derived with the LAGRangian ANalysis TOol (LAGRANTO). The SD mass is also computed from the size distribution time series for 5 years and extended to the 2001-2023 dataset. The 23-year long time series of SDEs at the JFJ allows to establish a long-term climatology of the dust frequency and mass in central Europe, and to study the seasonality and the inter annual variability of Saharan dust transport to the Alps. Finally, the source regions of particles are computed from FLEXPART near ground (< 40m) source sensitivities for SD detected by the optical properties and/or the coarse-mode particle concentration.



## 2 Experimental

### 2.1 Sites and instrumentation

#### 2.1.1 Jungfraujoch

The JFJ Sphinx observatory (46.5475 °N, 7.9792 °E, 3580 m a.s.l.) is a global Global Atmosphere Watch (GAW), an Aerosol, Clouds and Trace Gases Research Infrastructure (ACTRIS), and a Swiss National Air Pollution Monitoring Network (NABEL) station and has a longstanding record of aerosol observations (Baltensperger, 2024; Laj et al., 2024; Bukowiecki et al., 2016; Collaud Coen et al., 2020). It is located in a saddle between Mt Mönch and Mt Jungfrau in the Swiss Alps, which has a smooth slope along the Aletsch Glacier in the south-western direction and a rapid decrease in altitude towards north-east. The JFJ is usually sampling in the free troposphere in winter, but it is also regularly influenced by the atmospheric boundary layer (ABL) from spring to autumn. During the warmer months, convection over the plains and the thermal wind systems in the Alps can lead to the transport of ABL air masses to higher altitudes. A continuous aerosol layer, which stretches over the Alps, is consequently formed during fair-weather days and results in moderate aerosol concentrations at the JFJ (Nyeki et al., 2000; Collaud Coen et al., 2011; Herrmann et al., 2015; Brunner et al., 2022). The local convective boundary layer that develops in adjacent valleys in summer more rarely reaches the JFJ during the day, but then brings a higher aerosol concentration (Poltera et al., 2017).

In situ aerosol measurements at the JFJ are performed since 1995, but the spectral measurements of both the absorption and scattering coefficients are available only since March 2001. The availability of the instruments used for this study, CAMS dust products, FLEXPART source sensitivities and of the periods used for various comparisons are summarized in Fig. S1. The scattering coefficient has been measured by a TSI nephelometer since 1995 at 450, 550 and 700 nm, by an Ecotech Aurora 3000 from 1st of November 2013 until the 31st of July 2019 at 450, 525 and 635 nm and by an Airphoton N10 from 1st of December 2017 until May 2021 at 460, 525 and 631 nm. All the scattering coefficients were corrected for the truncation (Anderson and Ogren, 1998), but it was verified that the application of the truncation correction has negligible effect on the SDEs detection by the optical method. The spectral absorption coefficient was measured by filter-based absorption photometers, with an AE31 (@Magee Scientific) from the 21st of March 2001 until the 31st July 2019 and by an AE33 since the 15th October 2014, both measuring at seven wavelengths (370, 470, 520, 590, 660, 880 and 950 nm). Moreover a Multi-Angle Absorption Photometer (MAAP) provides the absorption coefficient at 637 nm with a higher accuracy since the 1st March 2003, so that the multiple scattering constant can be corrected for the high cross-sensitivity with scattering for high SSA as described by Yus-Díez et al. (2021). It was verified that this AE correction does not lead to quantitative better SDEs detection nor reduces the difference in SDEs detection between the TSI-AE31 and TSI-AE33. The size distributions of fine-mode and coarse-mode aerosol have been measured by a home-built Scanning Mobility Particle Size Spectrometer (SMPS) since 2008 (Wiedensohler et al., 2012; Jurányi et al., 2011) and by an white-light optical particle size spectrometer (FIDAS) since November 2016 (EMPA and BAFU, 2024), respectively. FIDAS measurements are calibrated through long-term comparison with the PM10 reference method (i.e. high volume filter gravimetry) and, thus, report data with respect to mean local temperature and pressure conditions. Surface





and volume of the aerosol were computed from the size distribution measured by both the SMPS and the FIDAS with a stich point between the two instruments at 450 nm. Aerosol mass density of 1.3 g cm$^{-3}$ (Cross et al., 2007) and of 2.65 g cm$^{-3}$ (Hess

et al., 1998) were used to convert the volume of aged continental aerosol and of SD, respectively. To estimate the SD mass, a 20 days moving median was applied to the size distribution excluding SDEs and attributed to background aged continental aerosol, whereas the remaining volume was attributed to dust.

### 2.1.2 Mount Helmos

The Mt Helmos station or Helmos Hellenic Atmospheric Aerosol and Climate Change station (HAC) (37.984° N, 22.1969° E,

2314 m a.s.l.) is located at the top of Helmos (or Aroania) mountain in the northern Peloponnese, Greece. It is a regional GAW station and provides data to ACTRIS Research Infrastructure. HAC is part of the time in the free troposphere (Foskinis et al., 2024) but is also largely influenced by the atmospheric boundary layer as well as by long-range transport of continental air masses, of Saharan dust and of biomass-burning aerosol (Foskinis et al., 2024). The scattering coefficient is measured by a TSI nephelometer 3563 and the absorption coefficient by both an AE31 (@Magee Scientific) and a Continuous Light Absorption

Photometer (CLAP) developed at the NOAA (Ogren et al., 2017) and measuring at 467, 528 and 622 nm. Aerosol is sampled through a PM10 cut-off inlet, while a Nafion dryer is used to keep RH below 40 % .

### 2.2 Detection of Saharan dust events

The methods for estimating the presence of dust at the sites are described below. They comprise two in situ measurements (the optical method and a detection by a higher concentration of coarse-mode aerosol), the detection of near-surface trajectories or

increased surface source sensitivities over the Saharan desert, and the estimation of the dust concentration by CAMS. All these SDEs detection methods are described thereafter and are based on hourly data.

### 2.2.1 The optical method based on the exponent of the SSA

The optical method is based on the inversion of the wavelength dependence of the SSA. It requires measurement of the scattering coefficient ($\sigma_{sp}$) and the absorption coefficient ($\sigma_{ap}$) at several wavelengths to compute the spectral single scattering

albedo ($\omega_0 = \sigma_{sp}/(\sigma_{sp} + \sigma_{ap})$). The wavelength dependence of the optical parameters is characterized by the Ångström exponent (AE, in particular SAE, AAE and SSAAE are the AE of $\sigma_{sp}$, $\sigma_{ap}$, and $\omega_0$), which is calculated by a linear fitting of the log-transformed data at all wavelengths. The absorption coefficients were evaluated at the nephelometer's wavelengths using the measured AAE values to compute the $\omega_0$ at three wavelengths and the SSAAE. It was verified that the use of pairwise wavelengths or of the shortest (370-520 nm) or longest (590-950 nm) AE wavelengths to detect SD leads to similar results

as the fitting procedure on all wavelengths. Usually, $\omega_0$ values decrease with longer wavelength, leading to positive SSAAE. Collaud Coen et al. (2004) showed that the negative SSAAE is representative of the presence of mineral dust in the JFJ and can be used as a near-real-time alert for dust events. The inverted sign of the $\omega_0$ spectral dependence is due to the large size of the mineral dust aerosol, which induces a wavelength (quasi-)independence of $\sigma_{sp}$ with SAE around zero or even negative, and the




presence of iron oxides in the Saharan dust, which increases their absorption of light at UV and blue wavelengths (300-400 nm)
leading to higher AAE ($\approx$1.5). The aerosol load is usually weak at high-altitude sites, so that nephelometers and absorption
photometers are regularly measuring near their detection limits, especially when sampling in the free troposphere. To avoid a
false SDE alert due to noisy signals in the JFJ, a latency period of 4 hours with a negative SSAAE (Collaud Coen et al., 2004)
was recommended to launch the SDE alert. Further minimal periods of at least 6, 12, 24 and 48 hours with SD are also used
for the comparison between different instruments or methods, as well as for the climatological analysis.

### 2.2.2 Method based on the in situ size distribution

Mineral dust contains a large amount of coarse particles up to 100 $\mu$m with mean geometric diameters around 2-10 $\mu$m (Ryder
et al., 2018; van der Does et al., 2018; Denjean et al., 2016). The FIDAS measures the aerosol size distribution of coarse
particles up to 25 $\mu$m at the JFJ. This method for detecting SD is based on a statistically significant (ss) increase in the number
of particles larger than 1 $\mu$m ($N_{1-25}$). To identify ss increases in $N_{1-25}$, a Kolmogorov-Zurbenko (KZ) low-pass filter (21-day
running mean) was applied three consecutive times to the time series. The statistical significance (ss) of the difference after
three iterations is determined following a normal law at the 95% confidence level, and is called the high frequency component.
It corresponds to periods with ss higher concentration of coarse-mode particles and can be used as a method to detect SDEs
when associated to back-trajectory analysis (Duchi et al., 2016). The JFJ data allowed us to verify that the choice of a confidence
level of 90% or 95% has a negligible impact on the detected SDEs (< 1% of SDEs hours). However, the KZ filter is sensitive
to the size of the moving average window with a decrease of $\approx$6% in the number of SD hours when applying a 21-d window
instead of the 19-d window recommended by Duchi et al. (2016). For the present study, a 21-d window and a confidence level
of 95% were applied.

### 2.2.3 Method based on CAMS' regional air quality analysis product

The regional air quality analysis products of the Copernicus Atmosphere Monitoring Service (CAMS) are based on an ensemble
of nine state-of-the-art numerical air quality models developed in Europe with a horizontal resolution of 0.1° (https://confluence.
ecmwf.int/display/CKB/CAMS+Regional%3A+European+air+quality+reanalyses+data+documentation, last access 12.08.2025).
O'Sullivan et al. (2020) found a good agreement between the CAMS dust product and the dust AOD observed by in situ and
remote sensing, as well as an underprediction of the coarse-mode dust associated with an overprediction of the fine-mode dust.
Hourly dust data were extracted for the closest grid point (46.55° N, 7.95° E) at 1000 m above the surface, corresponding to the
closest level to the JFJ real altitude in the CAMS modeled topography. CAMS dust product corresponds to a fraction in $PM_{10}$
and consequently predicts that dust is always present to some extent at the JFJ. Brunner et al. (2021) applied a threshold based
on the comparison between the SDEs detection by the optical method and the quartiles of the logarithm of the CAMS dust
concentration. To avoid cross-referencing between the detection methods, a KZ low-pass filter was also applied to determine
ss SDEs with the same parameters as for $N_{1-25}$ (3 consecutive iterations, 21-d period and 95% confidence level).





### 2.2.4 Method based on atmospheric transport modeling

FLEXPART is a Lagrangian transport and dispersion model suitable for the simulation of a wide range of atmospheric transport processes and includes dry and wet deposition, decay, and linear chemistry (Pisso et al., 2019; Bakels et al., 2024). It can be operated in forward and backward mode. In the latter it is similar to simulated air mass back-trajectories, but next to the mean transport also turbulent and convective mixing are considered. Here, FLEXPART was applied in backward mode for JFJ. Ever hour during the simulation period 20'000 model particles were released and followed backward for 30 days or until they left the domain of interest. FLEXPART was driven by analysis/forecast fields obtained from the operational HRES runs of the Integrated Forecast System (IFS) carried out by the European Centre for Medium Range Weather Forecast (ECMWF). FLEXPART outputs so called surface source sensitivities, which when convoluted with surface fluxes provide concentration enhancements at the receptor location. Here, source sensitivities were used in a qualitative way to detect general surface contact of air masses over the Saharan desert. Source sensitivities were generated for a sampling height of 40 m above ground, meaning that all model particles situated between the surface and 40 m are evaluated, whereas particles aloft are ignored in the output. Output resolution was 0.352° x 0.234°, which corresponds to about 25 km x 25 km over Europe, but 38 km x 25 km over the equator. Subsequently, the total source sensitivity over the Saharan desert (10°< latitude < 35° and -15° < longitude < 30°) was used as a SD detection method. Source sensitivities from the Sahara desert were frequently above zero. In order to detect SDEs, a KZ low-pass filter was also applied with the same parameters as for the coarse-mode particle concentration and CAMS dust concentration (3 consecutive iterations, 21-day period and 95% confidence level) to obtain ss SDEs based on FLEXPART source sensitivities. It was verified that the use of higher or lower confidence limits of 90% and 99% as well as of a source sensitivity minimum threshold has a negligible impact on the detected SDE hours.

The LAGRANTO COSMO, a Lagrangian parcel trajectories analysis tool, was used in the seven day forecast mode (Miltenberger et al., 2013) that are only available at 00:00 and 12:00 with a cell resolution of 0.5° corresponding to ≈ 35 km$^2$ over Europe. LAGRANTO was used in backward mode for the five grid points around the JFJ. The back-trajectories at 700 hPa were used to compute the number of hours over the Saharan desert without restriction of low-altitude back-trajectories as the sum of the five trajectories. Threshold values of 0, 50 or 100 hours were tested to represent potential SDEs, but the 0 threshold was finally chosen due to the very low common detection with other methods. It was also verified that the results were similar if three pressure levels (925, 700 and 540 hPa) instead of the solely pressure closest to the JFJ were used. To compare the LAGRANTO outputs with the hourly data for all of the other components, the values of the LAGRANTO back-trajectories were extended to -6 to +5 hours around the arrival time at the site. Another difference is that LAGRANTO is used in a forecasting mode, whereas both FLEXPART and CAMS are used in an analyzing mode.

Several studies (Banks et al., 2022; Lian et al., 2022; Varga et al., 2023) present dust events from deserts in the Middle East and in the Caspian/Aral regions. Applying the same methodology as for the Sahara desert, some LAGRANTO and FLEXPART back-trajectories effectively pass over these regions (latitude between 15.3 °N and 39.7 °N and longitude between 38.9 °E and 52.8 °E for the Middle East, latitude between 41 °N and 47 °N and longitude between 51 °E abd 63.5 °E for the Caspian/Aral regions) at low elevation. However, no dust event was observed with back-trajectories passing over these deserts in 2017-2021.





Obviously, information about surface wind speed or dust activation should be further used to better constrain the detection of
potential dust events (Zhang and Li, 2014), but this is beyond the scope of this study.

## 2.3 Results

### 2.3.1 Effect of the instruments' type to detect SDEs by the optical method

As presented in 2.2.1, the optical method for SDEs detection requires spectral observations of the scattering and absorption
coefficients. Since 2001, three types of nephelometers and two filters-based absorption photometers have been used at the
JFJ (Fig. S1). The five instruments were measuring simultaneously for one year, from August 1, 2018 to July 31, 2019. This
period is then used to compare the SDEs detection by the various pairs of instruments. Both absorption photometers (AE31 and
AE33) were produced by the same company (Magee Scientific), have a similar operational mode and use the same wavelengths.
However, the latest instrument (AE33) is corrected for variations in attenuation by using simultaneous measurements at two
filter spots with different loading (Drinovec et al., 2020) that leads to a better accuracy of the absolute value of the absorption
coefficients and of the spectral dependence of the loading compensation. The AAE distribution is presented in Fig. 1a for the
simultaneous measurement period, as well as for the first complete year of AE31 measurements in 2002. All AAE distributions
have a maximum around 1, but AE33 exhibits a more narrow distribution compared to AE31 and shorter tails on both sides.
With 17 years of operation, the left tail of the AE31 AAE distribution broadens between 0.6 and -4. The probability of having
a low AAE is higher for AE31 than for AE33, but the right tail used to detect SDEs remains similar.

The three wavelength nephelometers came from three different companies and have different characteristics such as the
illumination function, the zeroing algorithms and various wavelengths. The TSI nephelometer measures with a broader wave-
length range from 450 to 700 nm, whereas the Ecotech and Airphoton nephelometers only cover the range from 460 to 630
nm. The SAE distributions present some important differences between the instrument types (Fig.1b), which do not depend
on an applied minimal thresholds on the scattering coefficient (Figs. S2b and S3b). First, the TSI nephelometer has a SAE
maximal frequency of occurrence at 2.3, which is 0.5-0.7 higher than the Ecotech and Airphoton SAE ones, leading to a lower
probability for SAE to be smaller than 0.5. The Airphoton has a smaller left tail than the Ecotech, meaning a lower number of
very small and negative SAE. Consequently, the Ecotech has the highest probability to measure low SAE necessary for SDEs
detection. It should be noted that the TSI nephelometer has usually a unique lower maxima (result not shown) and that the
double maxima presented in Fig. 1b corresponds to a shift of the maxima during that year caused by calibration. Moreover, the
blue channel degraded with aging inducing higher noise affecting the precision since 2015 due to a gradual yellowing of the
light pipe and glue (Sheridan, 2020). The SAE distributions computed with only the blue and red wavelengths present similar
features (results not shown).

The resulting SSAAE depends on the pair of instruments used, and presents an almost symmetric distribution centered at
0.02, higher for TSI than for Ecotech or Airphoton (Fig. 1c). The SSAAE calculated with the TSI scattering coefficients has
also a 2-7 higher median and a twice higher variance than if calculated from the Airphoton or Ecotech scattering coefficients.
The SSAAE median and variance computed with AE31 are always higher than with AE33, whereas the kurtosis is smaller.





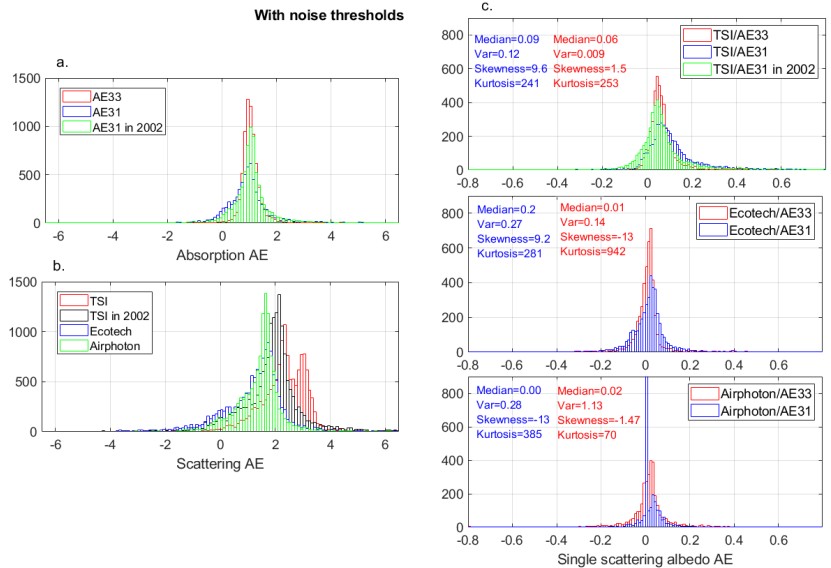

**Figure 1.** Histogram of the Ångström exponents for the simultaneous measurement period (August 2018 to July 2019) with the noise thresholds applied to the scattering and absorption coefficients for a) AAE for the AE31 and AE33, as well as AAE histogram for the first year of AE31 observation in 2002, b) SAE for the TSI, Ecotech and Airphoton nephelometers and c) SSAAE for all pairs of instruments

The higher variance of SSAAE if AE31 is used can be attributed to the higher noise induced by the aging of the instrument, whereas the higher SSAAE median and lower variance associated with the use of the TSI nephelometer can be explained by a spectral change due to the blue channel deterioration. All the SSAAE have low skewness and high kurtosis and resembles a normal distribution with longer tails.

The observed differences in SAE, AAE and SSAAE lead to large difference in the detected SDEs. Fig. 2a presents the various AEs during a long SDE in June 2019. The decrease in SAE values of about 1 unit for all nephelometers and the increase of about 0.5 in AAE are clearly visible from the 24th to the 27th of June 2019. On the 26th of June, AAE decreases to 1.2-1.0 and all SAEs increase, so that TSI SAE reaches 1.85 leading to a positive SSAAE and the end of the SDE detected by TSI-AE33. The lower Airphoton and Ecotech SAE values maintain negative SSAAE until the 28th of June at 07:00 and 11:00, respectively. If the SAE timeline of the three nephelometers are similar, the TSI SAE values leads to higher SSAAE and a much shorter reported SDE. The measurement of particle size distribution by the FIDAS (Fig. 2b) allows one to explain that the increase of SSAAE on the 26th of June is not primarily due to a decrease of the coarse-mode particles concentration but to an increase in the concentration of fine particles leading to a lower relative concentration of dust particles. The amount of coarse-mode particles tends to show that the SDE duration detected by the Ecotech and the Airphoton is reliable.

For the year of simultaneous measurements, the Ecotech-AE33 pair leads to the highest number of SD hours (Tab. 1). The discrepancy between the instruments is the highest in winter (Fig. S4), when low aerosol concentration leads to observations near the instrumental detection limits, as well as for the longest SDEs duration (> 12-24h). This last discrepancy occurs when





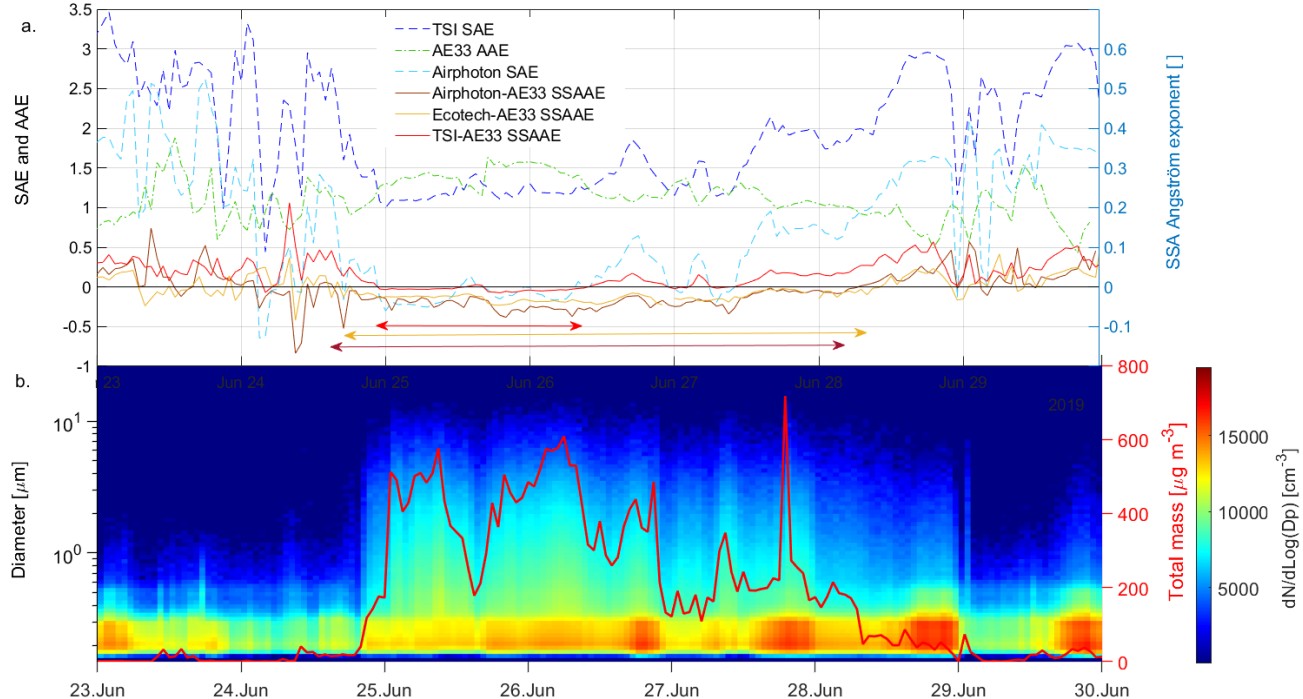

**Figure 2.** a) Scattering Ångström exponent from TSI, Ecotech and Airphoton nephelometers, Absorption Ångström exponent from AE33 and single scattering albedo Ångström exponent for TSI-AE33, Ecotech-AE33 and Airphoton-AE33 pairs for a SDE in June 2019. The colored arrows corresponds to the duration of the SDE reported by the various pairs of instruments. b) Particle size distribution measured by the OPC and total mass estimation by the SMPS and FIDAS size distributions.

the TSI-AE33 pair reports successive shortest periods instead of one long event due to e.g. spatial variability of the dust cloud

or ABL influence. Part of the discrepancy between the TSI-AE31 and the other instrumental pairs is explained by the aging of both instruments installed in 1997 and 2001, respectively. It has to be noted that Ecotech-AE33 also leads to a very high number of hours with SD at the high altitude station of Sonnblick (3106 m a.s.l..) in Austria (result not shown).

A similar comparison was done with one year (2022) of measurements at HAC involving a TSI nephelometer associated with either an AE31 or a CLAP providing the absorption coefficient at 467, 528 and 622 nm. HAC also samples part of the

time in the FT, but its aerosol concentration is globally higher with the 5[th], 25[th] and 50[th] percentiles of the absorption and scattering coefficients being 4 to 10 times larger than at JFJ. The number of SD hours detected by the TSI-AE31 or TSI-CLAP pairs are almost identical with less than 1.5% difference for each minimal SDE durations, leading to the conclusion that the nephelometer's type is probably a more crucial issue than the absorption photometer's type.



**Table 1.** Number of hours with SD detected by the various pairs of instruments for the simultaneous measuring period from 1[st] of August 2018 to the 31[st] of July 2019 with the noise thresholds as well as for SDEs detected by TSI-E31 or TSI-CLAP at HAC in 2022 without threshold.

| Nb hours with SD | $\geq$ 4h | $\geq$ 6h | $\geq$ 12h | $\geq$ 24h | $\geq$ 48h |
|---|---|---|---|---|---|
| JFJ Aug. 2018-Jul. 2019 | | | | | |
| TSI-AE31 | 116 | 99 | 67 | 0 | 0 |
| TSI-AE33 | 133 | 119 | 67 | 0 | 0 |
| Ecotech-AE31 | 771 | 888 | 447 | 115 | 52 |
| Ecotech-AE33 | 998 | 1378 | 807 | 309 | 111 |
| Airphoton-AE31 | 162 | 439 | 215 | 118 | 80 |
| Airphoton-AE33 | 694 | 867 | 370 | 72 | 0 |
| HAC, Jan. 2022-Dec. 2022 | | | | | |
| TSI-AE31 | 1014 | 921 | 752 | 608 | 268 |
| TSI-CLAP | 1028 | 916 | 749 | 605 | 268 |

### 2.3.2 SDEs detection at low aerosol concentrations

The aerosol concentration at high-altitude sites is very low in winter. The measurements are consequently performed at the detection limits of the instruments, which questions the reliability of the SDEs detection, especially for short events. The introduction of minimum thresholds in scattering and absorption coefficients to apply the optical method could then increase the confidence level of the SDEs detection. It was however verified that short SDEs measured during low aerosol concentration are sometimes confirmed by the back-trajectory analysis. The chosen threshold can be determined by the noise level of the

instruments or by the values of the optical parameters when SDEs estimated by the optical method are not confirmed by other methods. The AE33 median noises measured with filtered air were 0.05 and 0.01 Mm$^{-1}$ at 520 nm before and after the last official calibration, so that a noise threshold of 0.05 Mm$^{-1}$ is considered for the absorption coefficient. The TSI minute noise of 0.6 Mm$^{-1}$ at 550 nm is chosen as noise threshold for hourly scattering coefficient. More conservative scattering and absorption thresholds of 1 and 0.1 Mm$^{-1}$ are also tested to consider the median coefficients (1.28 and 0.11 Mm$^{-1}$ for scattering

and absorption coefficients) in case of SDEs detected by the optical method and not confirmed by the coarse-mode particle concentration and the FLEXPART source sensitivity. It was also verified that the values of the SSA and of all the optical Ångström exponents cannot be used to increase the confidence level of the SDEs detection. The introduction of thresholds in absorption and scattering largely decreases the variance and increases the skewness of the SSA (Figs. 1, S2 and S3), whereas the median values undergo slight changes of some percent's for SSA computed from TSI and Airphoton and larger modifications

of 30-50% if the Ecotech is used. The left and right tails of the SSA histograms are largely removed already by the lower noise thresholds . The number of SDEs (see Tables 1, S1 and S2) are largely decreased by the introduction of the higher conservative thresholds not only for short durations (factor of 2-2.5 for the pairs with the TSI, and 3-3.5 with the Ecotech and the Airphoton)





**Table 2.** Number of Saharan dust hours detected by the various methods in 2020 with the noise thresholds applied to the scattering and absorption coefficients.

| Nb hours with SD in 2020 | ≥ 4h | ≥ 6h | ≥ 12h | ≥ 24h | ≥ 48h |
|---|---|---|---|---|---|
| TSI-AE33 | 751 | 633 | 458 | 318 | 176 |
| Airphoton-AE33 | 1222 | 1072 | 590 | 375 | 231 |
| FIDAS | 1074 | 1003 | 813 | 524 | 153 |
| FIDAS+FLEXPART | 685 | 671 | 522 | 380 | 81 |
| CAMS | 1224 | 1198 | 1107 | 860 | 547 |
| FLEXPART | 1807 | 1759 | 1557 | 1255 | 956 |

but also for minimum durations of 12 h and 24 h for the Ecotech and Airphoton. The lower noise thresholds lead to similar but less pronounced effect. Finally the introduction of thresholds has negligible impact on the time and mass climatology. The

noise thresholds seem consequently more appropriate than the conservative ones, and will be used in the rest of this study.

### 2.3.3   Comparison with other SDEs detection methods

In the following, the optical method is compared to several other methods, such as the number of coarse-mode particles measured by the FIDAS, the amount of mineral dust modeled by CAMS, and the FLEXPART source sensitivity and the LAGRANTO back-trajectories passing over the Sahara. As explained in section 2.2.4, the LAGRANTO back-trajectories data

are prone to a much larger uncertainty due to their low time granularity and the use of the forecasting mode. The CAMS dust product is available since June 2019, so a different comparison period, year 2020, was chosen to compare the various SDEs detection methods (note that the number of SDEs and the magnitudes of the differences between the instrument pairs used for the optical method are substantially different in 2020 compared to 2018/19, but the general trends are the same). Tab. 2 shows that the FLEXPART method has the highest number of hours with SD for all the SDEs minimal durations. As already

presented in 2.3.1, the TSI-AE33 pair detects about 2/3 SD hours than the Airphoton-AE33 pair for short SDEs durations, whereas the differences between both pairs are smaller for the longest events. While the optical method with Airphoton-AE33, coarse-mode particle concentration and CAMS methods have similar SDEs hours for short SDEs durations, both the methods based on in situ observations lead to lower hours with SD for durations longer than 12 h. The histograms of the number of SDEs as a function of their duration (Figs. S6 and S7) are however similar for all detection methods so that the fewer long

lasting SDEs are only marginally compensated by a higher number of brief SDEs. It has to be noted that the coarse-mode particle concentration restricted to SDEs validated by the FLEXPART back-trajectory (FIDAS+FLEXPART) - corresponding to the method described in Duchi et al. (2016) - has the lowest number of SD hours for all minimal SDEs durations.

To further analyze the reasons for the discrepancy between the SDEs detection methods, Fig. 3 shows the number of hours with SD detected by all methods for each month between July 2019 and April 2021, and for the various minimal SDEs

durations. Missing data from any instrument are removed from all time series. SDEs detection by the optical method generally leads to more SD hours in winter, whereas SDEs detection by the coarse-mode particle concentration has more SD hours





in summer. CAMS dust product leads to similar results as the coarse-mode particle concentration. SD hours detected by the FLEXPART source sensitivities are generally more numerous but with a similar variability as if detected by the coarse-mode particle concentration and CAMS throughout the year; they are, however, smaller or equal to the SDEs hours detected by the

optical method in winter. The time series from 2017 to 2021 (Fig. S5) confirms that the coarse-mode particle concentration method leads to more hours from May to August and the optical method from November to February. The differences between both in situ methods can be explained in several ways. First, the higher number of SD hours detected by SSAAE in winter could be due to a higher rate of false detections due to the very low aerosol concentration at that time. Second, the higher number of SD hours identified by the coarse-mode particle concentration could be due to the presence of non coarse-mode particles

such as pollen, plant fragments, microplastics or local dust raised by agriculture and transported by ABL and mountain winds in spring and summer. This last assumption is supported by a higher accumulation mode in number concentration and size in summer (Rose et al., 2021; Herrmann et al., 2015). It was verified that periods with potential contamination from building work nearby the JFJ are not correlated to high concentration of coarse-mode aerosols. Finally, the optical method requires that dust dominates the aerosol optical properties, so that higher dust concentration is needed in summer, when the non-dust

aerosol concentration is enhanced by the entrainment of ABL air masses. Effectively, the number, surface, and volume of the particles as a function of the size distribution (Fig. S8) clearly shows that the concentration of coarse-mode particles is the highest for SDEs detected by both in-situ methods and the lowest for SDEs detected by SSAAE but not by the coarse-mode particle concentration.

Different methods to detect SDEs lead then to various SDEs influence at the JFJ as well a different seasonality. To better

understand the reliability and correlation between SDEs detection methods, parameters computed from the confusion matrix are evaluated. The confusion matrix provides the true positives (TP) and negatives (TN) as well as the false positives (FP) and negatives (FN). The accuracy measures the proximity between the two detection methods: $accuracy = (TP+FP)/(TP+FP+TN+FN)$. Precision gives the percentage of truly positive out of all positive predicted and characterizes the confidence to identify SDEs at the cost of some not detected events: $precision = TP/(TP+FP)$. Recall gives the percentage of predicted

positives out of total positives and characterizes the ability of the method to identify all SDEs at the cost of misidentified events: $recall = TP/(TP+FN)$. Finally, the F-measure is the harmonic mean of the precision and the recall and symmetrically characterizes both of them in one metric. Since we have imbalanced classes with rare occurrence of SDEs, the use of precision and recall are necessary indicators in the comparison of all SDEs detection methods. It should be noted that, when comparing two methods, the precision with one method taken as reference corresponds to the recall with the other method taken as

reference. Fig. 4 first shows that the accuracy always increases with the minimal SDEs duration, indicating that, as expected, the longest SDEs are detected well by all methods, whereas the shortest are more uncertain. The F-measure, the recall and to a lower extent the precision tend to decrease with SDEs duration for long SDEs. The best accuracy is found between the two in situ methods, the SSAAE and coarse-mode particle concentration methods, for events longer than one day. If the optical method is taken as a reference, the FLEXPART source sensitivity has the highest precision, and the coarse-mode particle

concentration the highest recall. This means that SDEs detected by the optical method are most of the time (45-85%) confirmed by FLEXPART, whereas it best matches the coarse-mode particle concentration. CAMS has a lower precision than FLEXPART



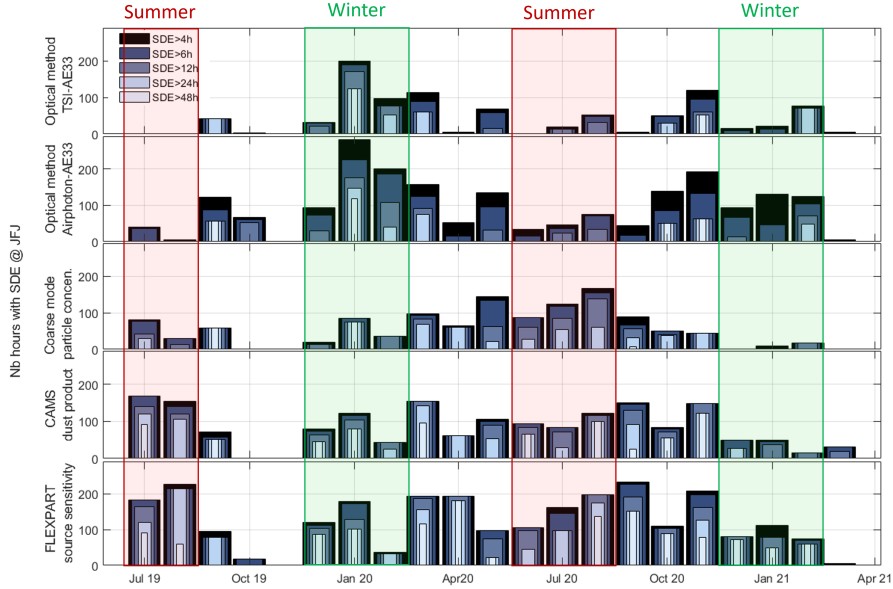

**Figure 3.** Number of hours with SD as a function of the minimal duration with the noise thresholds applied to the scattering and absorption coefficients for a) the optical method from TSI-AE33, b) the optical method from Airphoton-AE33, c) the coarse-mode particle concentration method from FIDAS, d) the dust index from CAMS and e) the method based on the FLEXPART source sensitivities. Summer and winter periods are colored red and green, respectively.

and a similar recall. The precision is low (< 0.4) for SDEs shorter than 12 h and increases with SDE duration for CAMS and FLEXPART, whereas it decreases for coarse-mode particle concentration and SDEs longer than 48 h. The recall and the F-measure are also very low (≤0.40) with the highest value found between both in situ methods. If the coarse-mode particle
concentration method is taken as a reference, CAMS has the best accuracy for short SDEs. The best precision, recall and F-measure are found for CAMS, which can be explained by the fact that both methods are based on a high amount of coarse-mode particles. Short SDEs lead to higher precision, recall and F-measure than long events. The FLEXPART source sensitivity also has high precision but a lower recall and F-measure than CAMS, and becomes less powerful than the optical method for SDEs longer than one day. The comparison between the model-based CAMS dust index and FLEXPART source sensitivities has a
low accuracy but a high F-measure due to similar intermediate values for precision and recall. This statistic can be explained by similar methods applied by models to compute the air mass transport, even if CAMS is specifically designed for coarse-mode particles. Finally, the parameters derived from the confusion matrix between LAGRANTO back-trajectories and FLEXPART source sensitivities have a low accuracy that increases with SDEs duration. However, the precision, recall and F-measure are almost constant with SDEs duration, even if the LAGRANTO time series is artificially reconstructed from the 12h and 00h
data.





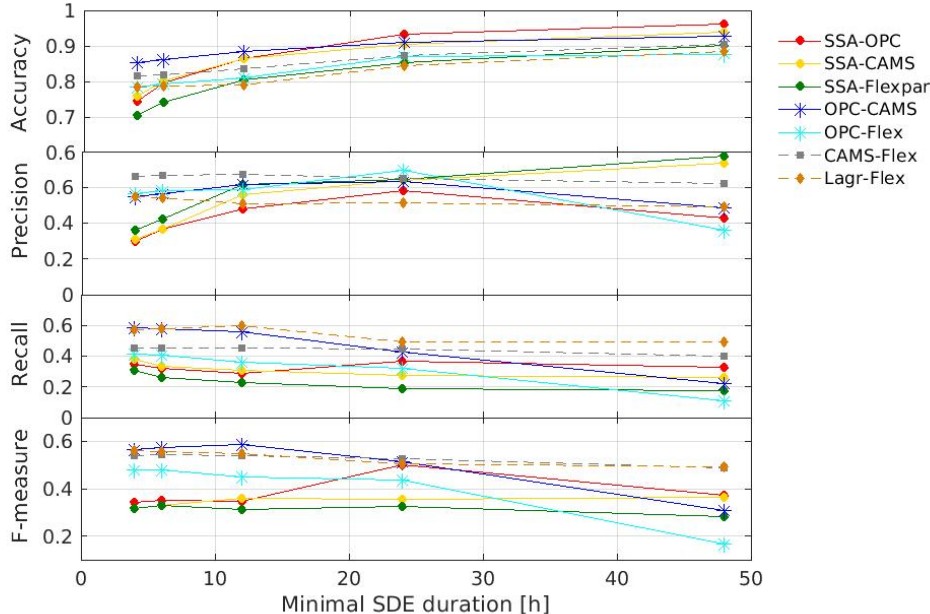

**Figure 4.** Accuracy, Precision, Recall and F-measure determined from the confusion matrix between pairs of methods to detect SDEs. The first mentioned method is considered as the reference method. The optical method is applied to the Airphoton-AE33 pair. It should be noted that the accuracy and F-measure does not depend on the chosen reference method, and that the precision and recall are inverted if the second mentioned method is taken as reference.

The results for the TSI-AE33 pair have accuracy, recall, and F-measure similar to the Airphoton-AE33 pair, but 0.1 lower precision compared to the coarse-mode particle concentration and FLEXPART source sensitivity, as well as for CAMS but only for short SDEs. These lower scores are in line with the decrease in sensitivity of the TSI to SDEs described above as a result of aging.

The accuracy calculated from the confusion matrix is generally high (>0.8) if the optical method is taken as a reference, the precision greater than 0.5 for long SDEs, but the recall is very low. The best performances are always due to measurements performed at the same location or detected based on similar type of observation. This comparison between the various SDEs detection methods leads to the main conclusion that none of the methods used can be considered as a reference for the SDEs detection at the JFJ due to their intrinsic detection criteria and artifacts. The optical method is very sensitive to low concentra-

tions of dust in a pristine environment, but is largely impacted by mixing with other aerosol sources and by noisy signal due to the weak aerosol load. The methods based on size distribution cannot differentiate SDEs from other coarse-mode aerosol such as biological particles, sea salt and local soil dust produced by site works, agriculture and traffic and an identification





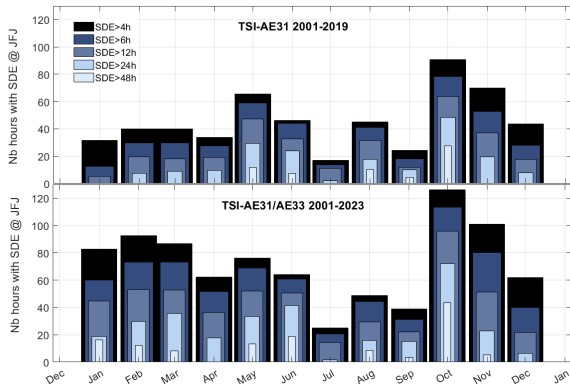

**Figure 5.** Mean monthly sum of the number of hours with SD detected by the optical methods with noise thresholds as a function of the minimal SDEs duration for a) 2001-2019 with the TSI-AE31 pair and b) 2001-2023 with the TSI-AE31 pair until 2014 and the TSI-AE33 pair since 2015.

could only be achieved by high time resolution measurements of the chemical composition. Finally, the modeled data contains uncertainties bounded to dust source and sink, and to poorly resolved terrain in complex topography.

## 2.4 Climatology of SD frequency and mass

A 23 year time series of SDEs detections is presently available at the JFJ since the beginning of the spectral measurements in 2001. Fig. 5 shows the climatology for 2001-2019 and 2001-2023 of the number of SD hours for different minimal SDEs durations computed based on the optical method from the longest time series with the same instruments (TSI-AE31) and from the combined TSI-AE31/AE33 pairs, respectively. Saharan dust has the strongest influence in fall (October-November), late spring (May), and in June and August based on the 2001-2019 period. These months also correspond to the one with the largest number of long SDEs lasting more than two days. July and January have the lowest number of hours with SD lasting at least 4 h and 6 h, respectively. As stated in sect. 2.3.1, the TSI-AE31 pair leads to the lowest SDEs detection. The SDEs climatology of 2001-2023 built from TSI-AE31 until 2014 and TSI-AE33 afterwards is very similar to that of 2001-2019. The main difference is the highest number of hours as well as longer SDEs in January-March due to strong SDEs in January 2020, February 2021 and March 2022. The strongest SDEs since 2001 effectively occurred in February 2021 and March 2022, leading to daily PM10 masses of 215 and 112 $\mu g\,m^{-3}$, as well as hourly maxima of 767 and 375 $\mu g\,m^{-3}$ at the JFJ, respectively. Both the 2001-2019 and 2001-2023 SDEs climatology refine the 2001-2002 one (Collaud Coen et al., 2004), which already described late spring and fall as the periods of the year with the strongest SD influence.

The SD mass was also evaluated (see section 2.1.1) for the January 2017 to April 2023 period when fine and coarse size distributions were both measured. The climatology of the dust hours (Fig. 6a) is generally similar to the 2001-2023 climatology if detected by the optical method. The climatology of dust frequency detected by the coarse-mode particle concentration (Fig. 6b) has more hours with dust than the optical method from May to September as well as in November and less hours from




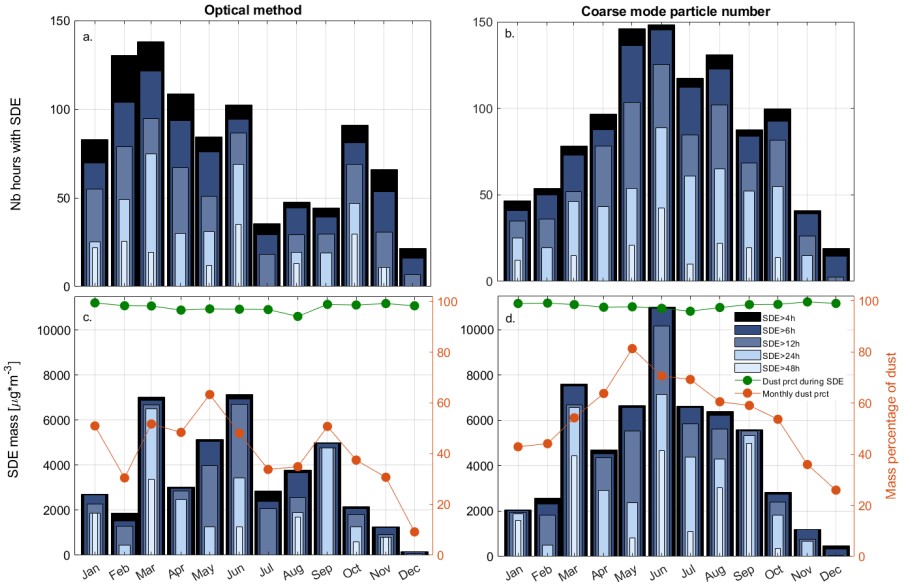

**Figure 6.** Climatology of SDEs hours and mass for the January 2017-April 2023 period for a) and c) detected by the optical method with the noise thresholds and b) and d) detected by the coarse-mode particle concentration method. The green dots corresponds to the dust mass percentage during the dust events and the red dots to the mean dust mass percentage during the month.

January to March as already described in subsection 2.3.3. The dust mass climatology values represent the total mass of SD per m$^3$ of air for each month, averaged over the years 2017-2023. This five year climatology shows that the dust mass detected

by the optical method is lower (100-7000 $\mu$g m$^{-3}$) than if detected by coarse-mode particle concentration (500-11000 $\mu$g m$^{-3}$). The differences are the larger from April to October when the ABL influence is the greatest, whereas the dust mass is similar from November to March. The mass percentage of dust during the events (green dots) reaches at least 94 % for both detection methods and is, as expected, the lowest if mixed with ABL particles in summer. If the optical method is used, the mean monthly dust percentage (red dots) is comprised between 30 % and 60 % for all months apart from December presenting only

9% . The dust mass percentage depends on the SDEs frequency as well as the mass contribution from ABL air masses and is consequently higher in spring and fall. The coarse-mode particle concentration leads to higher mean monthly mass percentage (50-80 %) between March and December, whereas the application of the Duchi method involving both the coarse-mode particle concentration and the back-trajectory method (Fig. S9) leads to a somewhat lower dust mass concentration and contribution. It has to be noted that the mass climatology is only affected in January by the introduction of noise or conservative thresholds to

restrict the scattering and absorption coefficients (Fig. S10).

    An Extinction to Mass Constant (EMC) of 3.8*10$^6$ (STD = 3.2*10$^4$) was computed from 2017-2023 time series with the conservative threshold of 1 Mm$^{-1}$ and SDEs lasting at least 12 h. The EMC for non-dust aerosol is of 5.2*10$^5$, namely more than 7 times smaller than during SDEs. These EMC were applied to the TSI-AE31 and the TSI-AE33 extinction coefficient at 550





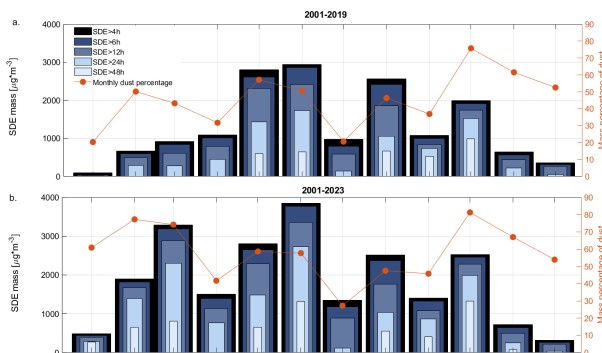

**Figure 7.** Climatology of dust mass and dust mass percentage detected by the optical method with the noise thresholds for a) the 2001-2019 TSI-AE31 time series and b) the 2001-2023 combined TSI-AE31 (2001-2014) and TSI-AE33 (2015-2023) time series.

nm to compute the dust mass and the mass percentage of dust from 2001 up to 2023. The 2001-2019 TSI-AE31 reconstructed

dust mass climatology shows dust masses between 1000 and 3000 $\mu$g m$^{-3}$ April to October apart from July, whereas the monthly mass percentage of dust is also higher than 50% in February,May, June, November and December when the dust load is important or when the JFJ is mainly in the free troposphere. The 2001-2023 climatology combining both AE31 and AE33 data is largely impacted by the intense SDEs in the first part of the year in 2020-2022 leading to high dust mass and/or mass percentage in January-March.

The monthly trend of SDEs frequency and mass as a function of the four meteorological seasons (Fig. 8) clearly emphases the high variability of the dust influence at the JFJ. The simultaneous measurement period of TSI-AE31 and TSI-AE33 pairs (January 2015-July 2019) illustrates that the differences between the instrument pairs is much smaller than the inter-annual variability. Marked variations of the SDEs frequency with time can be observed, in particular i) few winter events during the decade 2007-2017 and more frequent and more intense events since 2018, ii) few spring events during the decade 2010-2020,

iii) one or two long events per 5 years in summer, July presenting the lowest SD influence and iii) a more regular SDEs frequency in fall with very fewer events in September, particularly in 2009-2021. Taking only the TSI-AE33 measurements, the last four years (2020-2023) clearly present more SD hours at the JFJ than the previous five years (2015-2019). The highest monthly dust mass (Fig. 8b) and the monthly mass percentage of dust (Fig. S11) also occurs after 2019 for all seasons, but no obvious increase if the dust frequency and mass is visible. The evolution of the SD hours, SD mass and SD mass percentage can

presently not be considered as a long-term evolution. May-June and October-November are clearly the periods with often long and intense SDEs, but the inter-annual variability and the inhomogeneous SD time series at the JFJ prevent the determination of long-term trends in either the frequency or the intensity of the events.





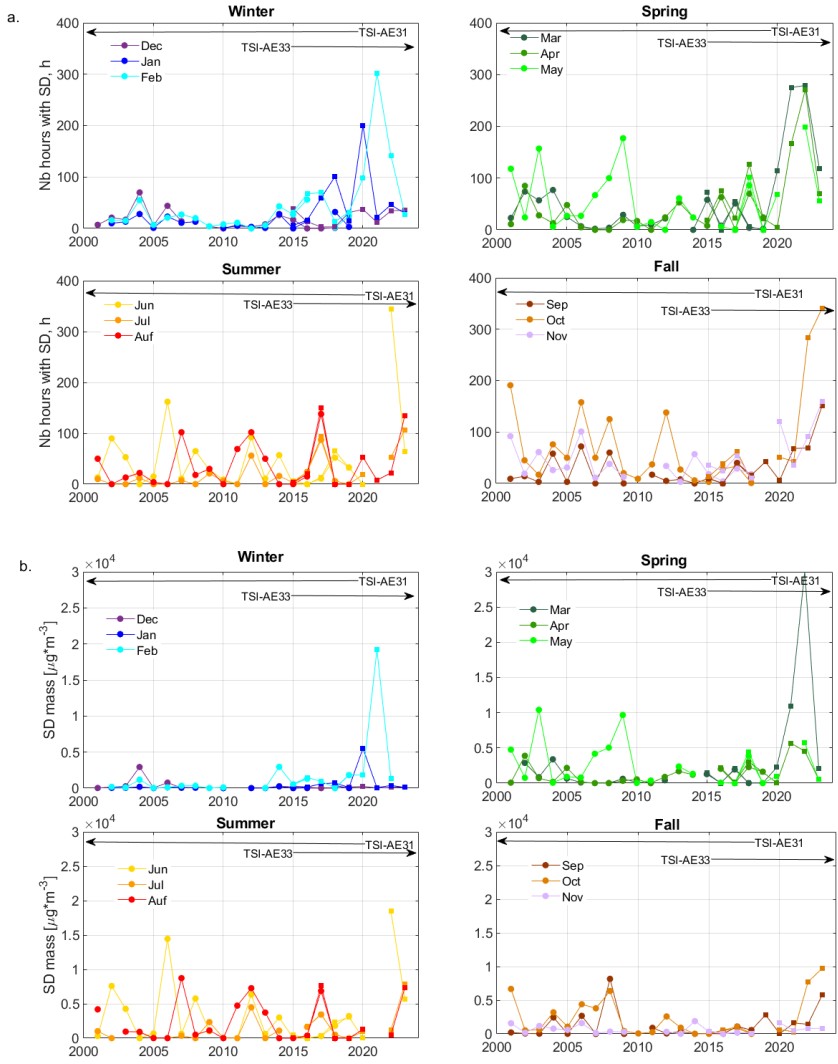

**Figure 8.** Monthly climatology of a) SD hours and b) SD mass deteted with the optical method with noise thresholds for the TSI-AE31 (squares) pair until July 2019 and the TSI-AE33 (circles) pair since January 2015 divided into the four seasons. Months with less than 50% of measurement are not reported.





## 2.5 SDEs footprints

The FLEXPART source sensitivity analysis allows to evidence the sources and transport pathways of air masses containing dust
before their arrival at JFJ. The source sensitivities of all SDEs (Fig. 9 a and c) point to the center, northern and northwestern
part of the Sahara as the main source regions with the highest density in south Algeria, Niger, Mali, Libya and Mauritania.
Some of the air masses also originate from the western Sahel belt. The relative source sensitivities (Fig. 9 b and d) present
clear differences as a function of the SD detection mode. First, the footprints of SDEs detected by both in-situ methods (Fig.
S12 b and c) target the whole Saharan region but also the Middle East, the Aegean Sea region, the Iberian Peninsula and the
Tyrrhenian sea, which can be either considered as potential source regions but also as travel ways. The source sensitivities
of SD detected by the the optical method (Fig. 9b) denote low relative values over southern Europe, so that a direct pathway
from the Sahara to the Alps seems to be favored with an homogeneous probability of path over the Mediterranean. On the
contrary, SD detected by the coarse-mode particle concentration presents higher relative footprints over eastern Mediterranean
and south eastern Europe as well as a more direct path to JFJ over the Iberian peninsula and south France. The relative source
sensitivities of events detected by the coarse-mode particle concentration but not confirmed by the optical method (Fig. S12 d
and e) clearly designate Spain and Turkey as main sources/path regions. Both these pathways lead to longer travel times over
land with a higher exposition to continental pollution, particularly grass, crop and forest fires, which are important in Turkey
and the Iberian peninsula in summer and could explain the high accumulation and coarse-modes associated with these events.

## 3 Discussion

### 3.1 Properties of the detections based on optical properties and size distribution

The SDEs detection by the spectral dependence of the optical parameters is widely used in both in situ and remote sensing
measurements. The multi-wavelength extinction of light by aerosol is an essential climate variable of the GCOS/WMO (WMO,
2025) so that the spectral behavior of the optical parameters is frequently used to characterize dust events. During dust events,
the in situ measured AAE is usually reported to be larger than 1.5 (Valentini et al., 2020; Schmeisser et al., 2017; López-
Caravaca et al., 2022; Yang et al., 2009; Collaud Coen et al., 2004) or even larger than 2 (Cappa et al., 2016; Costabile et al.,
2013). At the JFJ, 1.5 corresponds to the 75% percentile of the AAE's, whereas the median AAE values estimated from AE31
and AE33, 1.31 and 1.37 respectively, are somewhat smaller and similar to the value in Granada, Spain (Valenzuela et al.,
2015). The reported SAE values are often smaller than 0.5 (Valentini et al., 2020; Cappa et al., 2016; Costabile et al., 2013;
Schmeisser et al., 2017; Romano et al., 2019; López-Caravaca et al., 2022; Yang et al., 2009; Collaud Coen et al., 2004), but
also up to 1.1 in Granada, Spain. An even mean negative value of -0.6 was also reported at the Aitana peak, in Spain (López-
Caravaca et al., 2022). The median SAE value at the JFJ is about zero (-0.035 measured with the TSI) with inter-quartiles
ranges of 0.55 and -0.35. The published SSA values during dust events are more diverse (0.74 to 0.93) because of the different
mineral compositions as well as of the relative concentration of the BC and BrC in the various background environments. The
high altitude JFJ is a very clean site and has consequently a high median SSA of 0.91 during SDEs. Similar spectral behavior



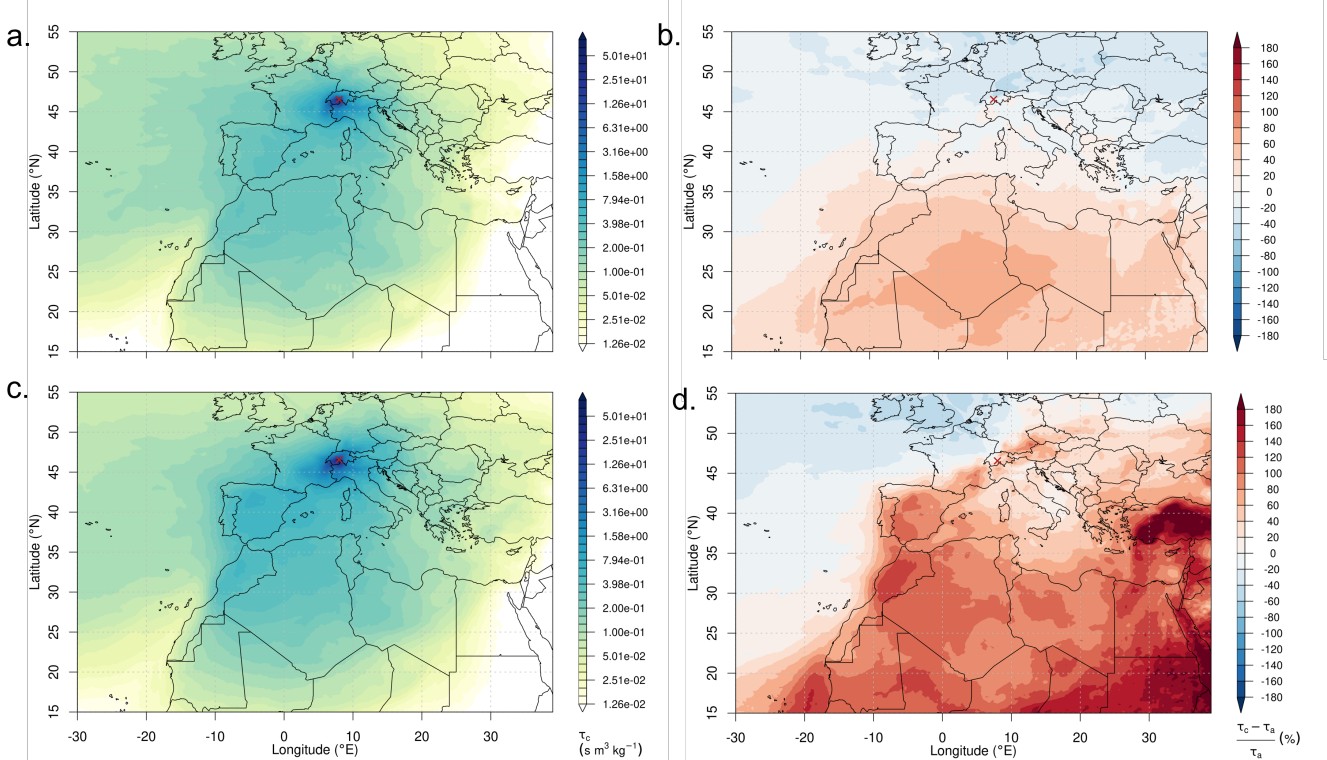

**Figure 9.** Average (a and c) and relative (b and d) source sensitivities based on 2018-2023 FLEXPART simulations during SDEs detected by the optical method (a and b) and coarse-mode particle concentration (c and d) for minimum duration of 4 h. The red cross corresponds to the JFJ. The average source sensitivities for the whole period, used to compute the relative source sensitivities are given in Fig. S12a.

of the columnar optical parameters are estimated from ground-based or satellite remote sensing measurements associated most of the time with retrieval algorithms to estimate the aerosol optical and physical properties. It has to be mentioned that retrieval algorithms uses assumptions (e.g. on homogeneity, lidar ratio, aerosol type, vertical distribution, diurnal cycles) to retrieve the SSA, the scattering, and the absorption as well as the Ångström exponents, leading to more homogeneous results than with in situ observations. The retrieved AAE and SAE are usually similar to the in situ observations (Valentini et al., 2020). The SSA

Ångström exponent is however rarely reported in studies using either in situ or remote sensing observations. A lot of figures or tables show its wavelength dependence or report SSA differences, e.g. dSSA=SSA($\lambda$ 2)-SSA($\lambda$ 1) with $\lambda$ 2 > $\lambda$ 1. Negative SSAAE and corresponding positive dSSA are found in case of aerosol dominated by mineral dust compound both in in situ (Valentini et al., 2020; Valenzuela et al., 2015; Kaskaoutis et al., 2021; Romano et al., 2019; Costabile et al., 2013; Russell et al., 2010) and remote sensing observations (e.g. in Russell et al. (2010); Valenzuela et al. (2015); Ge et al. (2010)). The

AAE versus SAE plot (Russell et al., 2010; Cazorla et al., 2013) relies on the effects of the aerosol size on SAE and on the effect of the chemical composition on AAE to classify aerosol into different types. Per definition, SSAAE is zero when AAE is equal to SAE and SSAAE is negative when AAE is larger than SAE under the condition that AAE is positive. This AAE larger





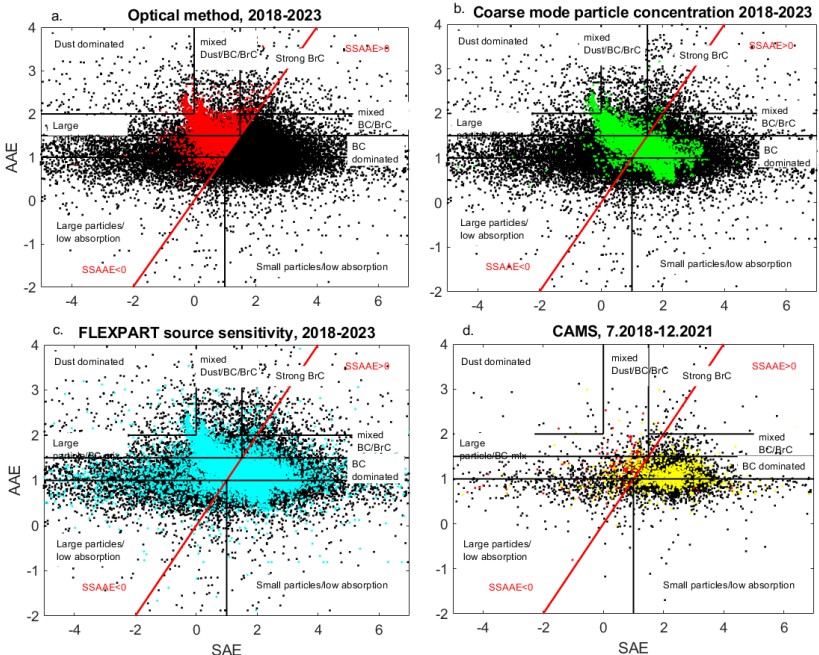

**Figure 10.** Scattering Ångström exponent as a function of the absorption Ångström exponent for 2015-2023 using TSI-AE33 pair for all data in black and for SDE detected by a) the optical method with noise thresholds, b) the coarse-mode particle concentration, c) the FLEXPART source sensitivity and d) the CAMS dust product from July 2018 to December 2021. The red line corresponds to the criteria used by the optical method.

than SAE condition is effectively reported during dust incursion from Africa and Asia (Schmeisser et al., 2017; Yang et al., 2009; Cazorla et al., 2013). The AAE versus SAE data are plotted in Fig. 10 for JFJ TSI-AE33 data for 2015-2023. The red

line corresponds to zero SSAAE and negative SSAAE can be observed in the classes defined as large non-absorbing aerosol, of pure mixed dust compounds, but also as strong BC or mixed BC/BrC classes. The large majority of SDEs corresponds however to pure and mixed dust or to a mix of large particles and BC. Some SD points lays in the strong BC category and a special case of biomass burning (BB) event with an accumulation mode centered at about 600 nm activate the SDE alert at the end of September 2023 (Masoom et al., 2025). It was however not possible to find another similar BB case with negative

SSAAE in the 23 years of measurements. This analysis suggested that the category "large particle/BC mix" contains a lot of dust events detected by all the methods described in this paper and should simply be merged in the mixed Dust/BC/BrC class. As expected the SDEs detected by the coarse-mode particle concentration but not the optical method correspond to either to a smaller mean diameter corresponding to higher SAE due to the mixing with aged continental aerosol or to lower absorption categorized mostly as BC dominated. SDEs detected by the FLEXPART source sensitivity and CAMS have equally dispersed

SAE and AAE.





The Ångström exponents depend on the used wavelengths. Most of the in situ and remote sensing instruments measure in the visible spectra (≈400-700 nm), some of them extending also in the near infrared and ultraviolet ranges allowing to study the spectral behavior from 350 to 1100 nm. Dust or mixed dust aerosol usually present a stronger wavelength dependency in the visible spectra than at higher wavelengths (Russell et al., 2010; Kaskaoutis et al., 2021; Valenzuela et al., 2015). The SDEs

JFJ time series even shows that the SSAAE computed from 450-550 nm is most of the time more negative than the SSAAE computed from 550-700 nm, but the use of the whole spectrum leads to less outliers. Some studies (Jordan et al., 2022; Valentini et al., 2020; Kaskaoutis et al., 2021) further used the spectral curvature of the logarithmically transformed parameters (e.g. extinction, AOD, AAE, SSA) to refine the differentiation of the aerosol species. The analysis of the curvature of the optical parameters at the JFJ as well as at several other remote sites (not shown) has not produced convincing results because

of either the low aerosol concentrations or the restricted number of wavelengths. Further types of representations such as SAE versus dSSA*AAE (Costabile et al., 2013) were proposed, but no indisputable method based on the wavelength dependence to detect the presence of dust in the atmosphere has been established. The optical method based on the negative SSAAE has the clear advantage to propose a clear threshold sensitive to both the size and the chemical composition of the particle. The attribution of periods with negative SSAAE to SDEs is not unequivocal. The risk of a false attribution of period with negative

SSAAE to the presence of mineral dust depends on the aerosol composition at the site. Since aged sea salt was never measured, the presence of large particles with low absorbing properties are improbable at the JFJ. BB events with big particles were measured but remain quite rare at the remote high-altitude stations, leading to a small and acceptable potential number of false attribution to SDEs. The mixing of dust with other aerosol species increases the difficulty to detect SD, particularly in case of urban sites situated within the ABL so that dust may no more dominate the aerosol optics. Moreover, the detection based on

the coarse-mode particle concentration can help identifying SD in case of BB events or strong mixing with accumulation mode particles.

## 3.2    SDEs detection and climatology at other sites

The SDEs climatology at the JFJ can be first compared with SDEs also detected by in situ aerosol measurements at four high- and middle-altitude stations, one north and three south of the Alps. Flentje et al. (2015) published the SDEs 1997-2013 clima-

tology at the Hohenpeissenberg (HPB) WMO/GAW station, a pre-Alpine hill at 980 m a.s.l. in Germany, Duchi et al. (2016) and Vogel et al. (2025) the 2002-2012 and 2003-2023 SDEs climatology at the Mt. Cimone (CMN) WMO/GAW global station situated at the highest peak (2165 m a.s.l.) in the northern Apennines in Italy, and Petroselli et al. (2024) the 2009-2018 SDEs climatology at Monte Martano (MM), an EMEP remote mid-altitude (1100 m a.s.l.) station in central Italy. The SD detection methods differed for the present study. A method based on the back-trajectories analysis and on the number of particles larger

than one micrometer that is ss higher than the background was used at CMN and MM, whereas a multi-factorial approach based on particle composition, particle volume distribution, optical properties and positive matrix factorization analysis allowed defining a Saharan dust index at HPB. A high occurrence of SDEs in spring is the main common feature of the decade climatology at all sites. Lower SDEs occurrences in winter are measured at the three northernmost stations (HPB, JFJ and CMN). JFJ in the middle of the Alps, HPB north of the Alps and MM in central Italy present a second maximal influence





during fall and more rare SDEs occurrence during summer. In contrast to HPB and JFJ, the stations south of the Alps have a stronger SD influence in November than in October. Finally, CMN used a method also based on the coarse-mode particle concentration and presents a larger SD influence in summer than in fall, similarly to the SD seasonality at JFJ detected by the same method. During the common analyzed period at HPB, CMN and JFJ (2002-2012), the years with the highest SD influence are even not completely similar at the three stations with shared maxima (e.g. in 2012 for the three stations, 2004 and 2006 for CMN and JFJ, 2003 and 2008 for HPB and JFJ) and minima (e.g. 2005 and 2010 for HPB and JFJ) but also with large differences. One of the strongest difference is a low SDEs fraction at CMN for the 2020-2023 period, whereas these four years corresponded to the highest SD hours measured in the JFJ timeseries. Globally, the SD influence in the Alpine stations (HPB and JFJ) is, however, more similar than at CMN. PM10 aerosol was also sampled and chemically analyzed for a one year period (August 2021-July 2022) at the Col Margherita Atmospheric Observatory (MRG, 2543 m a.s.l.) in the eastern Italian Alps (Barbaro et al., 2024). The main sources of PM10 were determined from source apportionment using positive matrix factorization. Most of the determined SDEs were also measured at the JFJ, but a lot of SDEs at the JFJ are not visible in the contribution of dust to PM10 mass. This is particularly the case for January, February and April 2022 with 123, 213 and 310 hours with SD at the JFJ. Cuevas-Agulló et al. (2024) analyzed the SD during winter 2003-2022 over the western Euro-Mediterranean region based on satellite aerosol products from MODIS/Aqua. Even if the vertical and horizontal scales were very different, most of the February and March events were also measured at the JFJ. The strong January 2020 event at the JFJ also induced high AOD over the western Europe. The comparison between SDEs at the JFJ and further SDEs climatology based on satellite observations over Hungary for the 2005-2021 period (Gammoudi et al., 2024), over the Carpathian Basin for the 1979-2018 period (Varga, 2020) and over Finland for the 1980-2022 period (Varga et al., 2023) shows that, if the intense events of February 2021 and March 2022 extended over the whole Europe, the northern and central European countries are usually not affected by the same dust events as in the Alps. For example, the SDEs climatology over the Carpathian Basin presents a high SD influence in spring and summer and only few events in fall and winter (Varga, 2020).

The very high inter-annual variability of the frequency and intensity of SD is observed not only at the JFJ (sect. 2.4) but also in all Europe (Petroselli et al., 2024; Cuevas-Agulló et al., 2024; Gammoudi et al., 2024; Varga et al., 2023; Varga, 2020; Duchi et al., 2016; Flentje et al., 2015) from observation of in situ and remote sensing, as well as from wet dust deposition in the Adriatic Sea (Mifka et al., 2022). The inter-annual variability is first explained by the variability of the emission rate (Mandija et al., 2018) and the atmospheric circulation (Gkikas et al., 2015). The SDEs variability is also influenced by the solar irradiance and the high latitude volcanic activity (Clifford et al., 2019). SDEs detection from glacier ice core proves that the high SD variability is observed not only in the last decades but even during the last 2000 years (Clifford et al., 2019).

The inhomogeneity of the SD time series at the JFJ restrains long-term trend analysis. Moreover, the length of the time series (23 years) is probably too short to analyze trends because of the high inter-annual variability of dust. While it is clear that the strongest SDEs of the last 20 years occurred in February 2021 and March 2022, no further assessments concerning the SDEs trends at the JFJ can be made. Some studies have analyzed long-term trend of SDEs from different types of observation and for various periods of time. Mandija et al. (2018) is based on a model associated to ground and space-based remote sensing observations; they found no trend in the SD emission rate and in the AOD over the plume area between 2006 and 2014. Light





decreasing dust AOD trends (max -5*10⁻⁵ yr⁻¹ per year) over the whole Europe were found for the 2007-2015 period at different altitudes, which are statistically significant only for the whole atmospheric column (Marinou et al., 2017). No trend in PM10 and PM2.5 associated with SD was found in Mt Martano, Italy, for the 2009-2018 period (Petroselli et al., 2024), whereas Pey et al. (2013) mention a decreasing tendency of SD influence in north west Mediterranean basin in the second half of 2001-2011. The analyses of thermodynamic variables bounded to SD transport to Europe predict an increase in the frequency of air

mass transport from North Africa to the western Mediterranean basin over 1948-2020 (Salvador et al., 2024). The global dust mass loading was shown to have increased by $55 \pm 30\%$ since the pre-industrial times (Kok et al., 2023), largely driven by increases in Asian and North African dust, whereas current climate models were not able to represent this historical increase. If an increase in intense SDEs during winter was observed in 2000-2022, they could also be attributed to the high inter-annual variability since no especially strong SDEs were observed during the three last three winters (2023-2025).

## 4 Conclusions

This study is based on 24-years of SDEs detected by the negative SSAAE at the JFJ extending from 2001 to 2023. Comparisons between three types of nephelometers and filter-based absorption photometers allow to assess the sensitivity of the optical method to the wavelengths range and their stability. A 4 years of SMPS and FIDAS time series allows a further characterization of the size distribution, coarse-mode particle concentration and fraction, a dust mass estimate as well as the mixing between dust

and aerosols from the atmospheric boundary layer. These dust detections were compared to CAMS dust products, FLEXPART source sensitivities and LAGRANTO back-trajectories passing over the Sahara desert.

A one year comparison between three nephelometers from TSI, Ecotech and Airphoton and three filter-based absorption photometers from Magee Scientific and the NOAA at the high altitude stations of JFJ and HAC showed that the type of absorption photometer is less critical than the type of nephelometer. In particular, SD detections involving a TSI nephelometer

and either an AE31 or a CLAP are completely similar at HAC, where the aerosol concentration is however higher than at JFJ. At the JFJ, the Aurora 3000 nephelometer leads to SD hours 6-10 and 1.2-1.5 times more numerous than the TSI and Airphoton nephelometers, respectively, as a function of the minimum SD duration and the used Aethalometer. Aging problems leading to a loss of sensitivity to dust are observed for both the 20 years old AE31 and the 27 years old TSI nephelometer.

Another year (2020) allowed the comparison between the various SD detection methods. The optical and coarse-mode

particle concentration methods present the lowest SD hours but there are clear disparities between both in situ methods with a higher detection in winter for the optical method and a higher detection in summer for the coarse-mode particle concentration. Dust detected by both in situ methods have the largest number concentration of particles with diameters bigger than 0.5 $\mu$m, but lower accumulation mode number concentration than if detected by the coarse-mode particle concentration alone. The exclusion of the scattering and absorption coefficients lower than the noise level largely decreases the short duration SD hours

but also impact the long events. CAMS has similar SD hours as the coarse-mode particle concentration for short minimum durations and more SD hours for events longer than one day. Finally, the FLEXPART source sensitivity method has the highest SD hours, which most likely relates to the absence of selecting only cases that showed favorable conditions for the activation



of dust in the studied source regions. None of the used methods can be considered as a reference method to detect SD so that the use of several methods together is recommended to increase the confidence level of the detection.

SDEs based on the optical methods are more frequent from February until June as well as in October and November. August presents sometimes strong events whereas December and January are usually affected only by short events. The mass climatology presents a similar pattern but lower dust mass in October and November. The inter-annual variability is very high leading to very different seasonal patterns. A clear increasing feature in SD hours is however visible since 2020 including the strongest SDEs measured in February 2021 and March 2022. The SD climatology based on the coarse-mode particle

concentration leads to largest influence from May to October, with numerous SD hours during summer contrarily to SD detected from the optical method. The SD mass detected by both the in situ methods always correspond to 94-100% of the total aerosol mass during the event. The climatology of the monthly SD mass percentage varies between 10% and 90%, but individual month with more than 80% of SD mass percentage can be find for all months of the year. Finally, the inhomogeneity of the SD time series, the large inter-annual variability and the relatively recent increase in SD hours restrains the determination

of trends in both the length and the intensity of Saharan dust at Jungfraujoch.

*Data availability.*  Aerosol in-situ data are available at the World Data Center for Aerosol (https://ebas.nilu.no/, last access 11.08.2025)

*Author contributions.*  MCC conceived the study, carried out the study and wrote the initial paper. BTB, MGB, RM and MS did the operational measurements at JFJ and MG and KE at HAC. SH computed the FLEXPART source sensitivity. BTB and DP contribute to the determination of the methodology. All co-authors revised the paper.

*Competing interests.*  The authors have no competing interests

*Acknowledgements.*  Copernicus Atmosphere Monitoring Service is acknowledged for providing the CAMS data. The authors thank Jesús Yus Díez and Cyril Brunner for fruitful discussions.

*Financial support.*  The continuous aerosol measurements at the Jungfraujoch site receive financial support from MeteoSwiss in the framework of the Swiss contribution (GAW-CH) to the Global Atmosphere Watch program of the World Meteorological Organization (WMO),

and from the Swiss State Secretariat for Education, Research and Innovation (SERI) in the framework of the Swiss contribution (ACTRIS-CH) to the ACTRIS ERIC. Empa's aerosol observations at the JFJ are part of the Swiss National Air Pollution Monitoring Network which is jointly run by Empa and the Swiss Federal Office for the Environment. FLEXPART simulations were generated as part of the project PARIS funded by the European Union under the Grant Agreement number 101081430. Measurements at HAC were partially supported by





the Action "National *N*etwork on Climate Change and its Impacts - Climpact", funded by the Public Investment Program of Greece, The
General Secretariat for Research and Innovation/Ministry of Development and Investments.



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
