# Peer review of "Detection and climatology of Saharan dust frequency and mass at the Jungfraujoch (3580 m asl, Switzerland)"

_EGUsphere, 2025_

## Author Response (AR1)

**Answers to the reviewer 1 comments on**

**"Detection and climatology of Saharan dust frequency and mass at the Jungfraujoch (3580 m asl, Switzerland)"**

*First, we would like to thank the reviewer for the valuable, in-depth comments on our manuscript. The answers to the comments and questions are written in italic thereafter.*

Overall Comments:

* It is not always clear in the text whether this refers to the complete data set or to data with applied noise thresholds (eg. L 225 ff, L 585 f)

*- The manuscript is effectively confusing since the noise and conservative thresholds are only described in section 2.3.2 SDEs detection at low aerosol concentrations whereas they are already used in section 2.3.1. Section 2.3.2 is consequently not at the right place in the manuscript and was moved as a new paragraph in section 2.2.1 describing the optical method. The fact that the noise thresholds are used for all the presented results is now introduced in the Experiment section, allowing us to clarify its usage in the Results and Discussion sections.*

*Answering this comment allowed also to correct the fact that the result section was a subsection of the experiment section. The numbering is now correct with 1. Introduction, 2. Experimental, 3. Results, 4. Discussion and 5. Conclusion.*

* The article focuses very much on Jungfraujoch data and the role of Mt. Helmos is not clear for the reader

*- A major part of this study is dedicated to evaluating and comparing the different methods commonly used to estimate the frequency of SDE occurrence and their contribution to PM mass. As part of this analysis, we assessed the sensitivity of SDE detection to different combinations of instruments measuring scattering (nephelometers) and absorption (aethalometers) coefficients. The sensitivity of SDE detection to different nephelometer types was directly evaluated at the Jungfraujoch, where a one -year comparison between three nephelometer was available. However, since only one type of absorption photometer was used at this*

*station, the inclusion of Mt Helmos data was necessary to assess SDE detection using two filter-based absorption photometers.*

*A comparison of the dust climatology at both sites would be very interesting, but we considered it more appropriate for a separate study, as a thorough analysis of several auxiliary parameters and meteorological conditions would be required to consider the unique characteristics of each site and accurately assess the SDE impact at each station. The length of the submitted manuscript prevents us from adding further content.*

* With regard to the noise threshold values, an evaluation should be carried out to rule out the possibility that this could introduce a certain influence or bias. Since the upper wavelength range (red or IR) will be the decisive criterion for the threshold value in both the nephelometer and the aethalometer (lowest scattering and absorption), a change in the AE also influences whether or not the noise threshold value is exceeded and hence, this might introduce a slight artifact.

*The chosen wavelength for the noise threshold is not the upper/red wavelength, but the green one in the middle of the spectrum (550 nm for the nephelometer measuring between 450 nm and 630/700 nm and 520 nm for the Aethalometer measuring between 370 nm and 950 nm). When the scattering or absorption coefficients at 550/520 nm are below the threshold, the data at all the wavelengths are no more considered for the SD detection. In that sense, no bias is introduced due to the use of only part of the wavelengths.*

Detailed                                                                                             suggestions:

L 109: as far as I know, AE31 measures at different wavelengths: 370, 450, 520, 590, 660, 880 und 950 nm

*The AE31 measures at the same seven wavelengths as the AE33, as specified in the sentence. The second wavelength is 470 nm and not 450 nm.*

Diagram 1: The differences in detail are very difficult to identify. For example, AE could be restricted to the range from -2 to 5.

*All plots of Figs. 1, S2 and S3 were restricted to smaller x-ranges to increase their readability.*

L 266f : The line implies that the hours of Saharan dust at Sonnblick are overestimated. There is no evidence to support this assumption.

*The researchers at Sonnblick limit the SDE detected by the optical properties based on an AE33 and an Ecotech by the PM10 mass. The results not shown here concern a comparison of SDE detected at JFJ, Sonnblick and Zugspitze during winter, when aerosol concentrations are very low, which indicates that the SDE frequency at SON is much higher than at ZUG and JFJ without any restriction applied. However, since these results are not presented in the paper, this sentence has now been removed from the manuscript.*

L 282 ff: which thresholds were used for Ecotech Nephelometer

*The same thresholds were used for all nephelometers and were always applied to the green channel (550 nm or 525 nm). The analysis of the SDE frequency (Figures and Tables of §2 and §4 of the supplemental material) clearly demonstrates that the differences in SDE frequency as a function of the nephelometer type are similar without thresholds applied to the scattering and absorption coefficients or with the noise or the conservative thresholds.*

L 416: space after comma between February and May

*Thanks, it is now corrected*

L 445 ff: As mentioned in the article, the Flexpart method also has its issues, and it could also be events that were detected using coarse-mode particle concentration but do not contain Saharan dust at all.

*Yes, our FLEXPART analysis does not take into account the meteorological conditions in the potential dust source areas (dust activation). Moreover, events with high concentrations of non-dust coarse-mode particles could be explained by the presence of bioaerosols, that are mostly released from March to September. The emissions and concentrations of bioaerosols are now referenced in the manuscript. The lack of chemical analysis and of aerosol typing does not allow us to identify such cases.*

*The manuscript was modified to clarify this point: "The relative source sensitivities of events detected by the coarse-mode particle concentration but not confirmed by the optical method (Fig. S12 d and e) clearly designate Spain, eastern Europe and Turkey as main sources/path regions. These pathways lead to longer travel times over land with a higher exposition to continental pollution, particularly grass, crop and forest fires, which are important in Turkey and the Iberian Peninsula in summer. Bio-aerosol larger than 1 µm comprises pollen, spores, plants debris and bacteria and their number and mass concentrations account typically for around 30\% in urban and rural air (Fröhlich-Nowoisky et al., 2016) with the highest European emission in the Iberian Peninsula, Turkey and Greece (Sesartic and Dallafior, 2011). Finally, the Anatolian plateau is also an arid region with potential*

*dust emission Hatzianastassiou et al., 2009; Aslanoglu et al., 2022). These events detected only by the coarse-mode particle concentration can then be explained by dust mixed with a high density of polluted accumulation mode aerosol impeding the detection by the optical method or by a high number of natural non-dust coarse-mode particles."*

L 571: 24 years or 23 years? I understand the difference between 24 calender years and 23 years time series but Ok, not important

*That's right, it was corrected to 23 years.*

**Answers to the reviewer 2 comments on**

**"Detection and climatology of Saharan dust frequency and mass at the Jungfraujoch (3580 m asl, Switzerland)"**

*First, we would like to thank the reviewer for the valuable, in-depth comments on our manuscript. The answers to the comments and questions are written in italic thereafter.*

1. **Lines 47 – 59:** I would suggest that the authors include a short discussion (2-3 sentences) acknowledging that the results may vary depending on the variable of interest, due to differences in their representativeness. This is attributed to the different representativeness of the selected variable. For example, aerosol optical depth (AOD) describes the total particle load throughout the atmospheric column, whereas ground-based in-situ measurements mainly reflect conditions within the planetary boundary layer. Consequently, elevated dust layers residing in the free troposphere may not be adequately captured by ground-based $PM_{10}$ measurements.

   *Yes, this is a good suggestion. The following sentences were added: "Ultimately, the various methods characterize various parts of the atmosphere. The representativeness of in situ measurements can be extended to the atmospheric boundary layer thickness in presence of a well-mixed atmosphere but is restricted to the surface layer otherwise. As column data, the aerosol optical depth detects the presence of dust without discriminating their altitude and consequently the potential effects such as*

*cirrus cloud or hail formation. The ground-based profiling techniques discriminate the altitude of dust from near surface to the free troposphere, leading to a higher potential for the validation of satellite and model data as well as for assimilation into models."*

2. **Line 65:** Are you referring to the application of the Ångström formula to the spectral single scattering albedos (SSAs)?

> *No, to our knowledge, the Ångström formula is bounded to the Ångström-Prescott model widely used for estimating global solar radiation. The notion of spectral curvature corresponds to the second derivative of the used parameter as a function of log($\lambda$). It can be computed by a second order fit (Kaskaoutis et al. 2021):*

$$\ln(abs) = A2 * \ln(\lambda)^2 + A1 * \ln(\lambda) + A0$$

> *Or simpler to compute it similarly to the Ångström exponent but with three wavelengths (Valentini et al., 2020):*

$$dAAE(\lambda 1, \lambda 2, \lambda 3) = 2 * \frac{AAE(\lambda 1, \lambda 2) - AAE(\lambda 2, \lambda 3)}{\ln\left(\frac{\lambda 3}{\lambda 1}\right)}$$

> *To clarify this point, the sentence was modified: ". Valentini et al. (2020) and Kaskaoutis et al. (2021) extended the use of optical properties to the absorption spectral curvature, relating to the second derivative as a function of the logarithm of the wavelength, and to the single scattering co-albedo Ångström exponent to further characterize the optical properties of various aerosol compounds."*

3. **Line 142 – 143:** What do you mean by "*The absorption coefficients were evaluated ….*"? Are you referring to the number of SDEs mentioned in the following sentence?

> *SSA has to be computed with the absorption and scattering coefficients at the same wavelengths. Due to the broader range of Aethalometer wavelengths (370-950 nm) than the Nephelometers' one, we chose to restrict the absorption to the Nephelometers' wavelengths. The sentence at line 142-143 means that the absorption coefficients were computed to the Nephelometers' wavelengths. We clarified the text as follows: "The Aethalometer covers a wider spectral range with finer resolution. Therefore, we used a power law to interpolate the spectral absorption coefficient data*

*to the wavelengths covered by the nephelometer before computing the SSA at these wavelengths and the SSAAE."*

4. **Line 152:** To classify a case as an SDE, is it required that all hourly SSA values within each of the defined time windows (e.g., 4 h, 6 h, etc.) be negative? Have you considered defining discrete time windows (e.g., 4–6 h, 6–12 h, 12–24 h, 24–48 h, and ≥48 h) to avoid overlapping?

   *Yes, we require that all hourly SSAAE (and not SSA) within the defined time windows must be negative. The choice to use minimal periods of time instead of discrete time windows was first used bounded to the SDE identification criteria of lasting at least 4 h in Collaud Coen et al., 2004. For various reasons bounded to instrumental problems, this minimal duration was set to 6h for a period. Finally, the publication of the SDE climatology in the MeteoSwiss annual climate report uses the quite intuitive representation like Figs. 3, 5, 6 and 7, that allow visualizing the number of monthly hours with dust as a function of their minimal durations.*

   *The number of SDE in discrete time windows can be deduced from the chosen representation. Both "methods" have pros and cons. As explained, our choice is somewhat historical, but we consider it as fit for purpose.*

5. **Sections 2.2.2 and 2.2.3:** It is not clear how the low-pass filter is applied. Is the first low-pass filter applied to the raw time series, the second to the already smoothed time series, and so on? How is defined the difference after the iterations?

   *The low-pass filter is applied as described in Duchi et al., 2016. Effectively the second/third low-pass filter is applied to the already smoother time series, so that the timeseries is smoothed three times. The difference between the original and the three times smoothed time series is considered as statistically significant by a normal low at the 95% confidence level. Practically, the normalized difference has to be larger than 1.96 times its standard deviation divided by the square root of number of valid data in the time series (N). In equations:*

   *data_diff= data-data_KZ3*

   *condition to be ss:*

   $$data\_diff - mean(data\_diff) > 1.96 * \frac{STD(data\_diff)}{\sqrt{N}}$$

   *The sentence was modified in order to clarify the procedure: "To identify ss increases in $N_{1-25}$, a Kolmogorov-Zurbenko (KZ) low-pass filter (21-day*

*running mean) was applied three consecutive times to the time series. The statistical significance (ss) of the difference between the original timeseries and the three times smoothed timeseries is then determined following a normal law at the 95\% confidence level and is called the high frequency component."*

6. **Section 2.2.3:** The CAMS ensemble is derived by nine or eleven models? (https://ads.atmosphere.copernicus.eu/datasets/cams-europe-air-quality-reanalyses?tab=overview). Which variable from the CAMS outputs is processed?

   *The CAMS ensemble is derived from eleven models since the 15 of July 2022. However, CAMS data for the years 2020 and 2021 are used, when the CAMS ensemble had only nine models. In order avoid adding non-critical details, we removed the reference to the specific number of models and rephrase it as "an ensemble of state-of-the-art numerical air quality models".*

   *The processed variable is the dust ensemble median from the CAMS Europe air quality forecasts. This dust variable was modified by July 2022 release.*

7. **Section 2.2.4:** I suggest clarifying more clearly why it is important for your study to utilize both FLEXPART and LAGRANTO-COSMO simulations. Why was FLEXPART run for such a long period (30 days)?

   *FLEXPART is a complete set of hourly data allowing a complete comparison with the in-situ and CAMS time series. The 30-day run was chosen to allow most particles to leave the model domain, which is useful for another application (greenhouse gas inversions) of the derived footprints. Particles that left the domain were terminated at the domain border and, hence, a shorter transport time was implied. For dust transport from the Sahara towards JFJ, transport times are usually in the order of a few days only and shorter integration times should have been sufficient. However, we don't expect the longer transport times to change the results.*

   *The comparison with Lagranto is useful since these data are operationally available at MeteoSwiss and could potentially be used for the operational SDE alert. In that sense, these results are necessary in case the SDE alert raised by MeteoSwiss will use these back trajectories.*

8. **Line 195:** Where is the KZ low-pass filter applied?

   *The KZ low-pass filter was applied to the total source sensitivity. It is now clearly mentioned in the manuscript.*

9. **Lines 317-320:** The agreement between the CAMS-based and coarse-mode-based SDEs is not very evident. Could the authors comment on the significantly higher number of SDE hours derived from the FLEXPART sensitivities compared to the other methods?

*The largest difference is clearly found between the optical method and the methods that are more sensitive to the coarse-mode concentration (coarse-mode particle concentration, CAMS and FLEXPART source sensitivity). The frequency and seasonality of CAMS is however the most similar to the coarse-mode particle concentration method. The sentence in line 317 was modified to specify these two points: "CAMS dust product leads to similar annual frequency and seasonality as the coarse-mode particle concentration."*

*As written in the conclusions in line L596, the FLEXPART source sensitivity method shows the highest SD hours, which most likely relates to the absence of selecting only cases that showed favorable conditions for the activation of dust in the studied source regions.*

10. **Figure 4:** I strongly recommend condensing the discussion section and emphasizing the main findings of the analysis. The current discussion is rather lengthy, which makes it difficult for the reader to clearly identify the key outcomes of this analysis.

*The description of Fig. 4 was shortened, and the emphasis is now on the interpretation of the results rather than on a simple comparison.*

11. **Lines 444-448:** I believe this part of the text requires further clarification. It is not clear how air masses traveling over regions where fine aerosols are more likely to be present can result in higher coarse-mode concentrations at JFJ.

*The formulation was effectively misleading and the manuscript was modified to add references on the number and mass concentrations of bio-aerosols and the potential emission of dust from the Anatolian Plateau : "The relative source sensitivities of events detected by the coarse-mode particle concentration but not confirmed by the optical method (Fig. S12 d and e) clearly designate Spain, eastern Europe and Turkey as main sources/path regions. These pathways lead to longer travel times over land with a higher exposition to continental pollution, particularly grass, crop and forest fires, which are important in Turkey and the Iberian Peninsula in summer. Bio-aerosol larger than 1 µm comprises pollen, spores, plants debris and bacteria and their number and mass concentrations account typically for around 30\% in urban and rural air (Fröhlich-Nowoisky et al., 2016) with the*

*highest European emission in the Iberian Peninsula, Turkey and Greece (Sesartic and Dallafior, 2011). Finally, the Anatolian plateau is also an arid region with potential dust emission Hatzianastassiou et al., 2009; Aslanoglu et al., 2022). These events detected only by the coarse-mode particle concentration can then be explained by dust mixed with a high density of polluted accumulation mode aerosol impeding the detection by the optical method or by a high number of natural non-dust coarse-mode particles."*

12. **Section 3.1:** I would suggest shortening this part of the manuscript to improve readability.

    *Section 3.1 was shortened by 14% and with the goal of improving the readability of the manuscript.*

13. The measurements acquired at Mt. Helmos should be given greater emphasis in the analysis. I would expect the authors to include an intercomparison of SDE characteristics (e.g., frequency of occurrence, intensity) between the two mountainous stations. Such an analysis would provide valuable insight, considering that JFJ and Mt. Helmos are influenced by dust outbreaks originating from different source regions and driven by different atmospheric circulation patterns.

    *The primary focus of this study was to compare and evaluate different methods for SDE detection, as they may yield quite different estimates of SDE frequency and their contribution to the aerosol load. Data from Mt Helmos was included in this paper to allow a comparison of SDE detected by the optical method using two different filter-based absorption photometers (AE and CLAP), which was not possible at the JFJ.*

    *The broad comparison of SD frequency and mass climatology across different high-altitude sites would be very valuable and should include not only these two stations but also further stations such as Mt Cimone, Sonnblick, Zugspitze, Pic du Midi, Moussala and Montsec. A first comparison between the three high altitude alpine stations (JFJ, ZUG and SON: https://www.meteosvizzera.admin.ch/servizi-e-pubblicazioni/pubblicazioni/rapporti-e-bollettini/2024/clima-delle-alpi-stato-del-clima-nelle-alpi-centrali-e-orientali-semestre-invernale-2023-24.html) showed that a quite extensive work on air-mass back-trajectory analysis is necessary to identify the arriving time of the dust plumes at the different stations. Meteorological conditions explaining the dust impact at each site should also be considered as part of such a study. Therefore, the comparison between the eastern mediterranean site of Mt Helmos and the*

*alpine site in the center of Europe was omitted as it would have result in a too long paper and is and beyond the scope of the present study.*